# Experience, circuit dynamics, and forebrain recruitment in larval zebrafish prey capture

Claire S Oldfield[1†], Irene Grossrubatscher[1†], Mario Chávez[2], Adam Hoagland[3], Alex R Huth[1], Elizabeth C Carroll[3], Andrew Prendergast[2,4,5], Tony Qu[3], Jack L Gallant[1,6], Claire Wyart[2,4,5]*, Ehud Y Isacoff[1,3,7]*

[1]Helen Wills Neuroscience Institute and Graduate Program, University of California Berkeley, Berkeley, United States; [2]CNRS-UMR, Paris, France; [3]Department of Molecular and Cell Biology, University of California Berkeley, Berkeley, United States; [4]INSERM UMRS, Paris, France; [5]Institut du Cerveau et de la Moelle épinière (ICM), Hôpital de la Pitié-Salpêtrière, Paris, France; [6]Department of Psychology, University of California, Berkeley, Berkeley, United States; [7]Bioscience Division, Lawrence Berkeley National Laboratory, Berkeley, United States

*For correspondence:
claire.wyart@icm-institute.org
(CW);
ehud@berkeley.edu (EYI)

†These authors contributed
equally to this work

Competing interest: See
page 24

Reviewing editor: Yuichi Iino,
University of Tokyo, Japan

**Abstract** Experience influences behavior, but little is known about how experience is encoded in the brain, and how changes in neural activity are implemented at a network level to improve performance. Here we investigate how differences in experience impact brain circuitry and behavior in larval zebrafish prey capture. We find that experience of live prey compared to inert food increases capture success by boosting capture initiation. In response to live prey, animals with and without prior experience of live prey show activity in visual areas (pretectum and optic tectum) and motor areas (cerebellum and hindbrain), with similar visual area retinotopic maps of prey position. However, prey-experienced animals more readily initiate capture in response to visual area activity and have greater visually-evoked activity in two forebrain areas: the telencephalon and habenula. Consequently, disruption of habenular neurons reduces capture performance in prey-experienced fish. Together, our results suggest that experience of prey strengthens prey-associated visual drive to the forebrain, and that this lowers the threshold for prey-associated visual activity to trigger activity in motor areas, thereby improving capture performance.

## Introduction

To transform sensory input into an optimal behavioral response, animals must extract relevant perceptual information from their environment, interpret it within their internal and external contexts, and translate it into a motor output. Prior experience modulates how this transformation occurs and whether the response is successful. A large body of work has studied how enriching or depriving sensory experience affects perceptual encoding, with both morphological and molecular changes (*Feldman, 2009*). Furthermore, teaching an animal to fear or expect a stimulus alters properties of the circuits recruited in response to the cue (e.g., *Letzkus et al., 2011*; *Matsumoto and Hikosaka, 2009*). Most studies of experience-dependent changes rely on drastic manipulation such as depriving animals of all sensory input in one modality, inducing fear association with a noxious stimulus, or depriving animals of food or water to achieve sufficient motivation to assure a response. However, neurons respond differently to ethologically-relevant stimuli (*Felsen and Dan, 2005*; *Theunissen and Elie, 2014*), and the question of how natural experience influences brain activity and downstream native behavior (*Sommerfeld and Holzman, 2019*) is becoming increasingly relevant.

One of the most critical native behaviors for survival in carnivores and omnivores is hunting for food. In many species, the basic hunting sequence is innate and triggered in full by certain sensory cues. For example, predation can be evoked in toads and fish by the sight of small prey-like moving objects (*Ewert et al., 2001*; *Bianco et al., 2011*; *Trivedi and Bollmann, 2013*; *Semmelhack et al., 2014*; *Matsunaga and Watanabe, 2012*) and in barn owls by ruffling prey-like noise (*Payne, 1971*). The accomplishment of this goal-directed behavior is highly flexible and is modulated by experience in animals as phylogenetically distant as mammals (the Etruscan shrew relies on tactile experience to develop efficient predation *Anjum and Brecht, 2012*) and mollusks (*Limax* learn to avoid a food if it makes them sick *Elliott, 2002*).

Here we investigated how experience-dependent circuit plasticity is implemented. We took advantage of the transparency of larval zebrafish and its ability to initiate prey capture when semi-immobilized, thereby making it possible to simultaneously image behavior and neural activity across a large portion of the brain. In zebrafish larvae, prey capture behavior is already evident at five days post-fertilization (dpf). At this stage, zebrafish respond to prey, such as paramecia, in a highly stereo-typed manner: when the prey is in sight, the fish reorients its body towards it with a series of unilateral tail flicks (J-Bends) and forward swims until the fish reaches a proximal striking zone; it then darts forward to engulf the prey in a final capture swim (*Borla et al., 2002*; *McElligott and O'Malley, 2005*; *McClenahan et al., 2012*; *Patterson et al., 2013*; *Johnson, 2019*; *Mearns et al., 2020*). Notably, the onset of this sequence is characterized by gradual eye convergence as the fish gets closer to the prey. The resulting increase of visual field area covered by binocular vision has been suggested to improve depth perception needed for precise targeting of the prey (*Bianco et al., 2011*). Restrained fish presented with virtual prey on a screen (a moving dot) also respond with eye convergences and tail flicks, indicating that visual inputs are sufficient to initiate the prey capture sequence (*Bianco et al., 2011*; *Trivedi and Bollmann, 2013*; *Semmelhack et al., 2014*; *Bianco and Engert, 2015*).

Prey capture in larval zebrafish has emerged as a model for understanding how sensory information translates into motor action (*Semmelhack et al., 2014*; *Bianco and Engert, 2015*; *Gahtan, 2005*; *Smear et al., 2007*; *Del Bene et al., 2010*; *Fajardo et al., 2013*; *Muto and Kawakami, 2013*). Visual information about prey location flows from the retina to two contralateral visual areas: the pretectum and optic tectum (OT). The pretectal area around the 7th arborization field of retinal ganglion cells (AF7, see *Burrill and Easter, 1994*) was shown to be critical for detecting prey-like objects and triggering the prey capture sequence (*Semmelhack et al., 2014*; *Antinucci et al., 2019*). Ablation and optogenetic studies indicate that the OT is necessary for prey capture (*Gahtan, 2005*; *Del Bene et al., 2010*). Assemblies of medial peri-ventricular neurons in the OT activate prior to eye convergence, suggesting a role in inducing the motor response to the sight of prey (*Bianco and Engert, 2015*). Furthermore, novel results have shown that the nucleus isthmi, a small cholinergic area in the cerebellum, is necessary for maintenance of a hunting routine, but not for initiation (*Henriques et al., 2019*). Despite recent progress, the mechanism for integration of information from pretectum and OT, and the precise activation sequence downstream of visual areas, are yet to be discovered. Moreover, it is not known whether prey capture improves with experience, and if so, how improvement might be implemented at the neural level.

Here we show that experience of live prey increases capture initiation and success in natural conditions. We investigate the brain activity elicited by prey and identify sequential activity in visual areas (pretectum and optic tectum), and motor-related areas (cerebellum and hindbrain). We find that prior experience of prey increases the reliability of capture initiation in response to prey-associated visual activity. In prey-naïve and prey-experienced animals, information flow from the pretectum onto the cerebellum and hindbrain correlate with prey capture initiation. However, experience of prey increases the impact of these functional links on prey capture initiation and strengthens the coupling from visual areas to the telencephalon. In agreement with this latter point, prey-experienced animals show increased activation of the forebrain (both telencephalon and habenula) during prey capture initiation. Consistently, we show that ablation of the habenula reduces hunting in prey-experienced animals. Taken together, our findings show that prey capture behavior is enhanced by prior experience of live prey and suggest that forebrain recruitment increases output gain for the prey capture circuit in response to the same visual cues.

## Results

### Prior experience of prey increases prey capture initiation in larval zebrafish

To assess the effect of experience on prey capture behavior and the underlying neural activity, we compared two groups of sibling zebrafish larvae. Prior to being tested with paramecia at 7 dpf, a first group was fed live paramecia for two days (5 and 6 dpf), whereas a second group was fed inert food flakes (*Figure 1A*). While each group may acquire feeding experience for the food on which it was 'trained,' only the first group obtained experience of live prey, on which both groups were later tested. We therefore refer to the first group as 'prey-experienced' and the second group as 'prey-naïve'.

At day seven prey capture behavior was tested in both groups by quantifying behavioral steps of the prey capture sequence: (a) pursuits that are aborted before a capture swim is attempted, (b) capture swim attempts that fail, and (c) successful captures (*Figure 1A* bottom and *Figure 1—video 1*). We found that experienced fish have significantly more pursuits and successful captures than their naïve counterparts, but the same probability of attempting a capture once a pursuit was initiated (*Figure 1B*). Experience did not change the probability of success once a capture was attempted. This analysis suggests that experience increases initiation of prey capture, but not motor performance of the capture.

We next examined prey capture in a virtual environment (*Bianco et al., 2011*; *Trivedi and Bollmann, 2013*; *Semmelhack et al., 2014*; *Bianco and Engert, 2015*) in prey-experienced or prey-naïve fish. We presented a single moving dot of varying contrast to a fish immobilized in agar with eyes and tail free (*Figure 1C*). We focused on initiation of prey capture by examining frequency of eye convergences, as described above. We quantified performance using the discriminability index, d', calculated from response rates to a stimulus *versus* the absence of a stimulus (see Materials and methods, Behavioral data analysis and statistics). We found that at higher contrast, prey-experienced fish responded significantly more than prey-naïve fish (*Figure 1D–E*). This indicates that response to 'virtual prey' is substantially enhanced by prior experience of live prey, even though the virtual prey is a black dot moving steadily and unidirectionally on a white screen, whereas paramecia are translucent and swim erratically in three dimensions.

If differences in diet (live paramecia *versus* inert flakes) affected fish health, there could be an effect on prey capture performance. To address this concern, we carefully examined numerous indices of fish health. We found that fish length, spontaneous swimming velocity, and swimming velocity in the presence of prey did not differ between the two groups of fish (*Figure 1—figure supplement 1A*). Similarly, swim distance and rest times between swims (*Figure 1—figure supplement 1E and F*), as well as rates of baseline tail flicks, eye saccades and eye convergences in the virtual environment did not differ between groups (*Figure 1—figure supplement 1B*). These results indicate that differences in diet between the two groups of fish did not affect health, and that the improved capture of paramecia results from prior experience of prey.

A difference in prey capture for experienced and naïve fish could also emerge if there was a difference in hunger or motivation to hunt. To disentangle the effect of experience on feeding from hunger, we performed a flake feeding assay. We fed sibling fish either paramecia or flakes on day 5 and 6 dpf ad libitum, and then starved the fish overnight, as in our usual protocol (*Figure 1A*). On day seven, we let fish feed on flakes for 10 min. We assessed the number of eye convergences during that time as a proxy for the motivation to hunt. We observed a significantly higher number of eye convergences in response to the flakes in previously flakes-fed fish, suggesting that flake-fed fish are no less hungry/motivated and that the experience with flakes boosts flake 'hunting' in the same way as experience with paramecia boosts paramecia hunting (*Figure 1—figure supplement 1G*).

### Spatio-temporal brain activity pattern associated with prey- capture initiation

To understand the neuronal basis of the boost in prey capture initiation in prey-experienced fish, we imaged neuronal activity in the form of calcium transients in *Tg(NeuroD:GCaMP6f)* (*Rupprecht et al., 2016*) transgenic zebrafish, which expressed the calcium indicator GCaMP6f

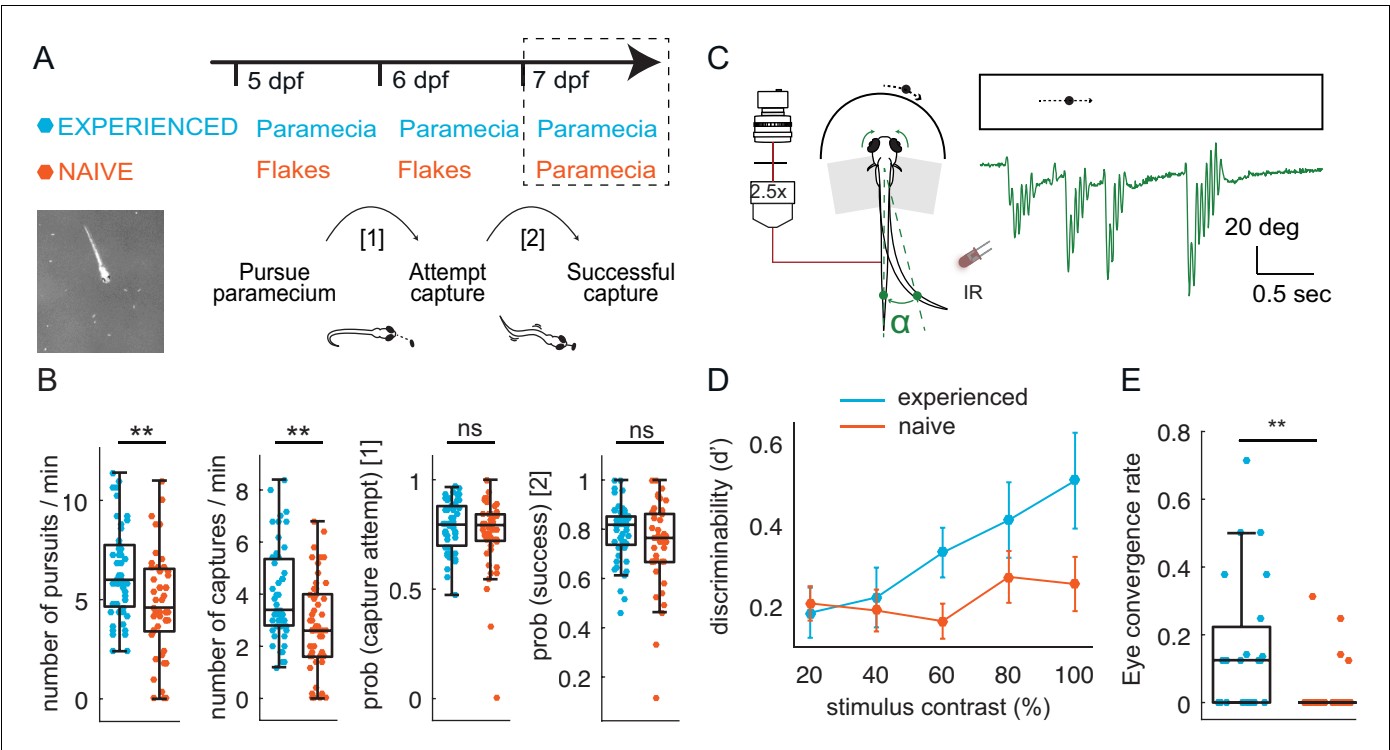

**Figure 1.** Larval zebrafish improve hunting performance with experience of live prey in both free- swimming and virtual environments. (A–B) Experience of live prey increases frequency of paramecia captures in a freely swimming environment. (A) Behavioral paradigm: Fish fed paramecia ('prey-experienced') or flakes ('prey-naïve') at 5 and 6 dpf were given paramecia at 7 dpf (top, timeline). Prey capture performance was assessed by imaging single fish and paramecia (white specks in lower left image) to count pursuits aborted without a capture attempt, failed capture attempts, and successful captures (summary behavior scheme, lower, right). (B) Summary of performance. Raw data (one symbol per fish) and a boxplot of group statistics show that experienced fish have higher frequencies of total pursuits (successful or not, p = 0.003), and successful captures (p = 0.001), but statistically indistinguishable probabilities of transitioning from pursuit to a capture attempt (p = 0.28), or of transitioning from capture attempt to successful capture (p = 0.12). Statistical comparisons used a permutation test (see Materials and methods) with N = 51 each experienced and naïve fish. (C–E) Experience of live prey increases frequencies of prey- capture initiation in semi-immobilized fish. (C) Setup: semi-immobilized fish face a screen on which small moving dots are projected. Tail flicks and eye angle are imaged from above at 250 fps. Alpha is the angle between the point at 8/10ths of tail length from swim bladder, and midline. In green we show an example tail track during presentation of moving dot. (D) Prey-experienced fish (N = 23) have significantly (p = 0.03) greater discriminability index (d') than prey-naïve fish (N = 25), two-way ANOVA interaction between experience of live prey vs. lack thereof and contrast (see Materials and methods for calculation of d'). (E) At highest contrast, eye convergence rate in prey-experienced fish was significantly (p = 0.005) greater than in prey-naïve fish (# of times fish converged eyes / # of high contrast stimuli at highest contrast). Note high variability in response rate within groups, with experience improving virtual prey capture performance unevenly across fish, similar to *Trivedi and Bollmann, 2013*; *Semmelhack et al., 2014*. See also movies *Figure 1—video 1* and *Figure 1—video 2*. Data tables for panels B, D and E in *Figure 1—source data 1*.

The online version of this article includes the following video, source data, and figure supplement(s) for figure 1:

**Source data 1.**
**Figure supplement 1.** Lack of impact of other factors on prey capture performance.
**Figure supplement 1—source data 1.**
**Figure 1—video 1.** Free-swimming prey capture.
https://elifesciences.org/articles/56619#fig1video1
**Figure 1—video 2.** Virtual prey capture.
https://elifesciences.org/articles/56619#fig1video2

broadly in the central nervous system. Imaging was performed in 7 and 8 dpf fish which were semi-immobilized in agar but with eyes and tail free, similar to the assay with virtual prey above. We monitored fluorescent calcium signals from a single plane that included the pretectal area around AF7 (see *Figure 2—figure supplement 1A*), which was previously shown to be involved in prey detection (*Semmelhack et al., 2014*) and in triggering the prey capture sequence (*Antinucci et al., 2019*), as well as the OT, motor-related areas, and other areas of the hindbrain and forebrain. We

simultaneously imaged eye and tail movements, as well as the trajectory of a paramecium that swam freely in a slot-well in front of the fish (*Figure 2A–C*). For each fish, we used the baseline GCaMP6f fluorescence image (*Figure 2D*) to identify the major brain areas (*Figure 2E*). Consistent with our assays on semi-immobilized fish above (*Figure 1D and E*), tail flicks were similar between prey-experienced and prey-naïve fish, but prey-experienced fish exhibited significantly more eye convergences than prey-naïve fish (*Figure 2F*).

We began by recording spontaneous neural activity and associated tail and eye movements in the absence of prey for a period of seven minutes. We then added a single paramecium to a well in front of the fish and recorded for 11 additional minutes. We focused on brain activity associated with both spontaneous and prey-evoked eye convergence events, time-locked to the moment of strongest ocular vergence. In prey-experienced fish these events showed strong activation of visual areas (pretectum and OT) and motor-related areas (cerebellum and hindbrain) (*Figure 2G and H*), that have been shown before to play roles in swim behavior and prey capture (*Antinucci et al., 2019*; *Henriques et al., 2019*; *Dunn et al., 2016*; *Ahrens et al., 2012*). The spatio-temporal patterns of spontaneous and evoked activity were similar, except that activity was more asymmetric in the pretectum and tectal neuropil in presence of prey (i.e. more strongly activated contralateral to the paramecium) (*Figure 2—figure supplement 1B*), in agreement with a moving stimulus being present in one hemifield and not the other.

We observed responses in AF7 in the pretectum and in the rostral neuropil and periventricular neurons of the OT (output neurons *Scott, 2009*), but not in the caudal neuropil (*Figure 2G*). Regionalization of activity in the tectal neuropil is consistent with prey capture initiation occurring when the paramecium is in front of the fish, because, under these conditions, the prey is in the nasal visual field, and nasal retinal ganglion cells project to the rostral optic tectum (*Karlstrom et al., 1997*).

Prey-naïve fish also responded to the sight of prey with eye convergences and tail flicks, albeit at a lower frequency than prey-experienced fish (*Figure 2F*). The spatio-temporal pattern of brain activity associated with eye convergence was similar to what we observed in prey-experienced fish for both spontaneous- and prey-evoked events (*Figure 2—figure supplement 1B*).

In addition to the activity expected in visual and motor-related areas, eye convergence-associated responses were observed in areas of the forebrain not previously implicated in prey capture: the telencephalon and habenula (*Figure 2B, G and H*, *Figure 2—figure supplement 1C and D*). We return to analyze this forebrain activity later.

## Experience of prey does not affect encoding of prey position in visual areas

To determine whether experience of prey affects the ability of fish to detect and represent prey location, we compared encoding of prey location in visual areas of prey-experienced and prey-naïve fish (*Figure 3*). Traditionally, visual responses are evaluated by repeatedly showing identical virtual stimuli, pooling trials and determining if responses in a given region of interest are reliable enough for it to be deemed 'visually-responsive'. However, natural stimuli have a more complex statistical structure (*Simoncelli and Olshausen, 2001*), and are thought to evoke more ethologically-relevant behavioral responses (*Wu et al., 2006*). We therefore used natural visual input by presenting the fish with its biological prey, a live paramecium. Since locomotion of the paramecium is not experimentally-controlled, we faced the analytic challenge of dealing with irregular visual stimulation. To address this, we applied a method developed for building predictive encoding models of human brain activity elicited by natural scenes or language, and detected by neurophysiology and functional magnetic resonance imaging (*Nishimoto et al., 2011*; *Huth et al., 2016*). We used regularized regression to construct a separate encoding model for each pixel that predicts the pixel's fluorescence time series based on the location of the prey (*Figure 3A–B*). To validate the encoding models, we predicted fluorescence time series on held-out segments of the dataset that were not used for model estimation, and then computed the correlation between predicted and actual time series in this held-out set.

For pixels where prediction performance was significantly above chance, we computed the prey position that elicits the largest response and defined it as the pixel's 'preferred angle' (*Figure 3A–C*). The encoding model weights describe the spatial receptive field of the pixel (*Figure 3C*, see Materials and methods and *Figure 3—figure supplement 1A*). Neurons in visual areas preferred positions on the contralateral side of the animal, consistent with retinal ganglion cell projections

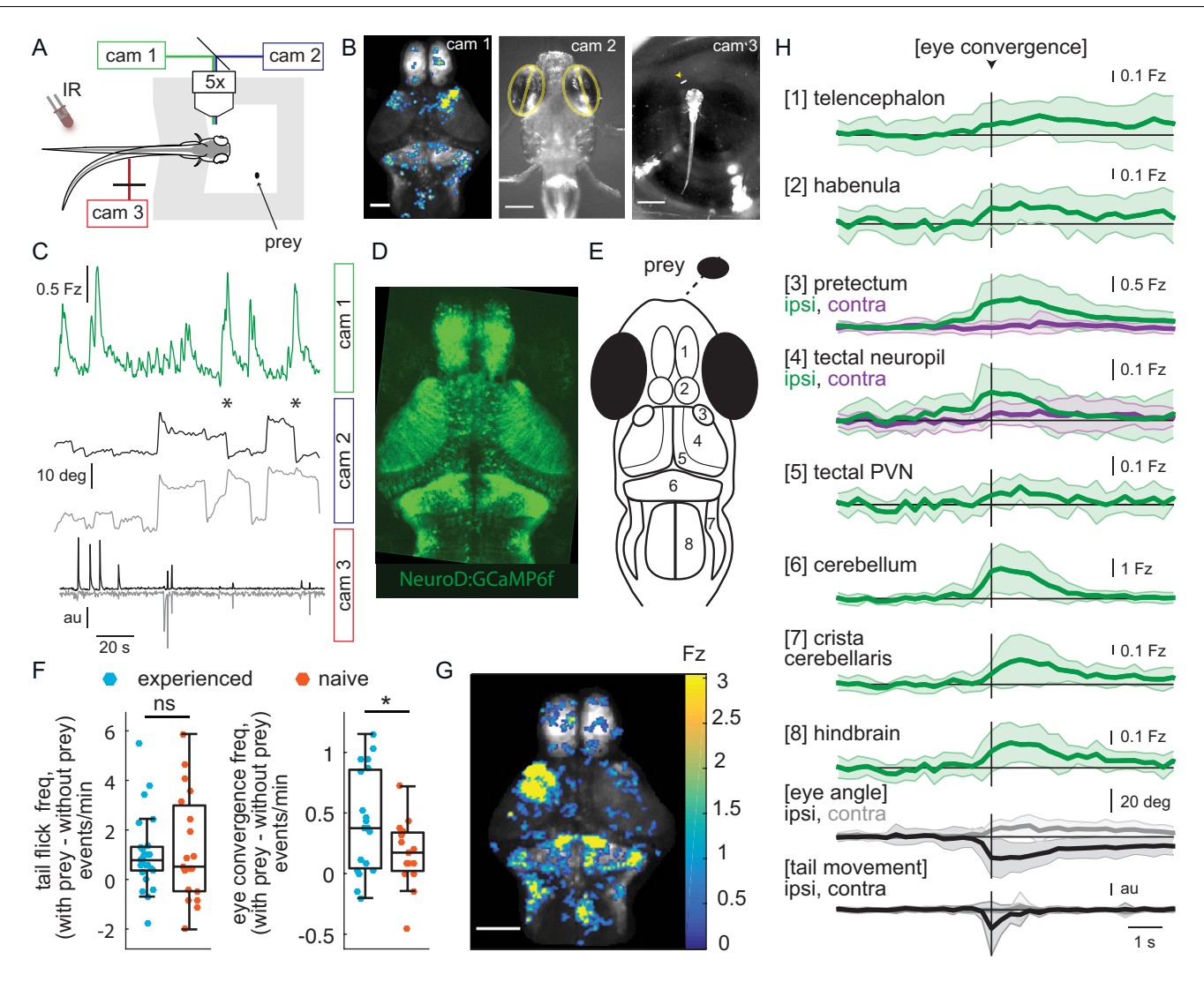

**Figure 2.** Wide-field brain imaging of prey capture initiation shows recruitment of visual and motor areas, as well as the telencephalon and habenula. (A, B) Setup for imaging of neural activity in a single plane of the whole brain while the fish observes prey (A) and example frames captured by three cameras (B). Camera 1 (cam 1): neural activity in a single plane of the whole brain while the fish observes prey, scale bar = 200 µm. Camera 2 (cam 2): eye angle, scale bar = 200 µm. Camera 3 (cam 3): prey position and fish tail position, scale bar = 1 mm. Cameras were synchronized at 3.6 Hz. (C) Example 3 min traces from one fish for all three cameras illustrating data collected during eye convergences. Cam 1: Z-scored fluorescence in the right pretectum (smoothed with a Lowess filter, span = 7, for Fz calculation see Materials and methods, Calcium and behavior imaging data pre-processing). Cam2: Corresponding eye angles (left eye, gray; right eye, black; convergence events, stars; smoothed with a Lowess filter, span = 9). Cam 3: tail movement (left side, gray; right side, black, see Materials and methods). (D) *Tg(NeuroD:GCaMP6f)* 7 dpf fish brain, dorsal view as imaged by cam 1. (E) Schematic of anatomy in observation plane, numbered areas as defined in (H). (F) Prey-experienced and prey-naïve fish have statistically indistinguishable evoked (with prey – without prey) frequency of tail flicks (left, p = 0.74), but prey-experienced fish have a significantly higher eye convergence frequency (right, with prey – without prey, p = 0.04). (G, H) Neural activity in a prey-experienced fish around eye convergences with prey (N = 12 eye convergences). This fish showed no spontaneous eye convergences preceding addition of paramecium, suggesting that averaged activity was purely evoked by the paramecium. 'Contra' and 'ipsi' refer to the side with higher or lower pretectal transient amplitude peak time (see Materials and methods). (G) Spatial distribution of summed calcium activity over 4.2 s (five frames before to 10 frames after eye convergence), when the prey was to the right side of the fish (average of six convergences). Scale bar = 100 µm. Fz thresholded for visualization. (H) Time-course of calcium activity for each brain area in an example experienced fish (average of 12 convergences; convergence time is vertical black line) over a period of 10 seconds. We observe a significant increase in fluorescence for all brain areas except the ipsilateral side of the pretectum, comparing average fluorescence traces of baseline (frames −10 to −5 before eye convergence), to the five frames after eye convergence (black line in the figure). P-values are reported in *Supplementary file 1*. For eye angle and tail movement, black is contralateral, and gray is ipsilateral. See also *Figure 2—figure supplement 1*, and movies in *Figure 2—video 1* and *Figure 2—video 2*. A permutation test was used for all pairwise comparisons if not specified otherwise (see Materials and methods, Behavioral data analysis and statistics). Data tables for panels F and H in *Figure 2—source data 1*.

*Figure 2 continued on next page*

*Figure 2 continued*

The online version of this article includes the following video, source data, and figure supplement(s) for figure 2:

**Source data 1.**
**Figure supplement 1.** GCaMP6f expression pattern and time traces.
**Figure supplement 1—source data 1.**
**Figure 2—video 1.** Movie showing simultaneous whole-brain activity and behavior imaging.
https://elifesciences.org/articles/56619#fig2video1
**Figure 2—video 2.** Movie showing spatio-temporal pattern of brain activation associated with eye convergence.
https://elifesciences.org/articles/56619#fig2video2

crossing the midline and innervating contralateral visual areas. We observed strong retinotopic gradients in both the pretectum and the optic tectum, with more rostral pretectal and optic tectum locations preferring central positions of the prey and more caudal pretectal and optic tectum locations responding preferring lateral positions of the prey. Retinotopic maps were equally well-defined in prey-experienced and prey-naïve fish, with both groups of fish showing well-separated bimodal distributions for angle preferences between left and right sides (*Figure 3D and F*). We observed no difference in encoding strength between prey-experienced and prey-naïve fish in the pretectum, tectal neuropil or tectal periventricular neurons (PVNs, *Figure 3E*, Materials and methods, Pixel-wise encoding model estimation and validation, for calculation of pixel correlation). The mean and standard deviation of preferred angle for each area were also similar between prey-experienced and prey-naïve fish (*Figure 3—figure supplement 1B*), indicating similar tuning characteristics. These results indicate that prior experience of prey does not affect encoding of prey position in visual areas.

In an additional analysis, we asked whether the threshold for a visual response to prey differed between animals that did or did not have prior experience of prey. We focused on the pretectum because this is the first relay for visual information coming from the retina and because the pretectal area around AF7 has been suggested to be specifically involved in prey detection (*Semmelhack et al., 2014*). Indeed, we found that in both prey-experienced and prey-naïve fish, pretectal pixels had higher correlation values with prey position than the OT (*Figure 3E*), for prey-experienced fish, the 25th and 75th percentile for average pixel correlation values were 0.02 and 0.09 respectively, and for prey-naïve fish 0.02 and 0.07 respectively. In addition, we observed no difference in frequency or amplitude of calcium transients in the pretectum when fish were observing a prey between prey-experienced and prey-naïve fish (*Figure 3—figure supplement 1C*). These results indicate that experience does not affect the threshold of prey detection within visual areas.

## Information transfer in visual areas during prey observation

Having observed that frequency of prey capture initiation is augmented by prior experience of prey, but that neural responses to prey in visual areas appear not to depend on experience, we asked whether activity in other brain regions or communication between brain regions differs in prey-experienced animals. To address this, we applied Granger-causality analysis, a method for determining if a time series of events predicts (or Granger-'causes') a second time series (*Friston, 1994*; *Granger, 1969*). It has been classically applied to neurophysiological recordings in both animals and humans (*Seth et al., 2015*; *Fallani et al., 2015*) to study the influence of one brain area or neuron upon another (*Figure 4A*, and see Materials and methods, Granger-causality from calcium fluorescence imaging data). In contrast to standard correlation analysis, Granger-causality provides directionality information (compare *Figure 4* and *Figure 4—figure supplement 1*), although it does not determine whether an apparent functional connection corresponds to direct or indirect anatomical connections, or whether it is serial rather than triggered in parallel by other brain areas. It should be pointed out, that the Granger-causality calculation relies on calcium activity dynamics captured at 3.6 Hz but elicited by action potentials that occur on a much faster time scale (although bursts of action potentials may occur over time scales that more closely resemble calcium dynamics). As a result, two regions that are in appearance functionally connected might seemingly peak at the same time because of slow calcium dynamics and/or image acquisition.

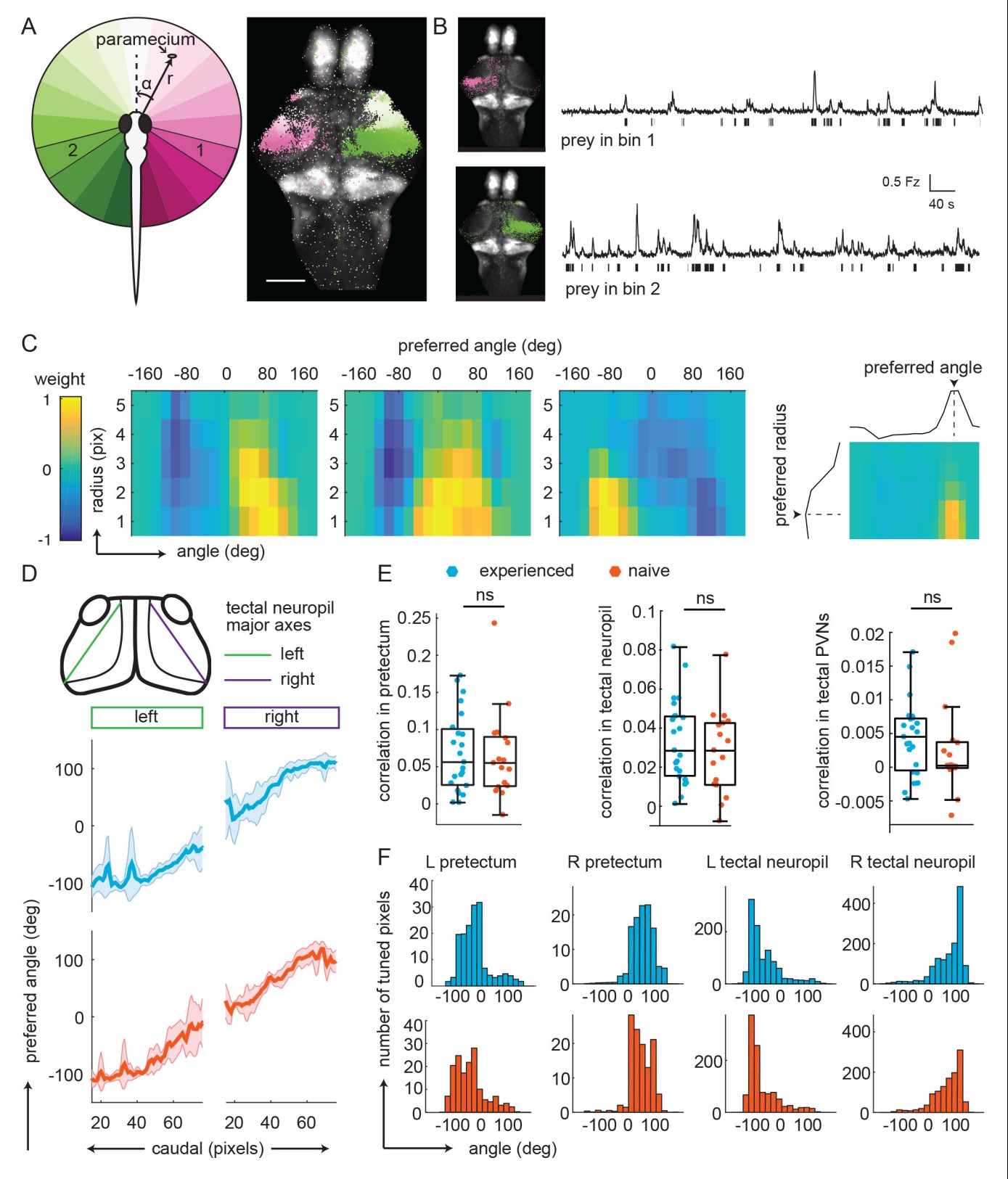

**Figure 3.** Experience does not affect prey-associated activity in visual areas. (**A**) Left, schematic of prey location relative to the fish in polar coordinates (angle α, radius r). Right, example retinotopic map generated by fitting an encoding model for each pixel to predict fluorescence intensity based on prey location. Significantly correlated pixels are in the color of their preferred angle. Scale bar = 200 µm. (**B**) Average fluorescence from pixels whose

*Figure 3 continued on next page*

*Figure 3 continued*

preferred angles are in bin 1 (120° to 101°, top) or bin 2 (-104° to 126°, bottom). Left: Anatomical location of pixels. Bars below traces indicate time points when the prey was present in the preferred angle bin. (C) Example angular-radial receptive fields for three pixels in the pretectum. X-axis: angle, y-axis radius; Color represents encoding model weight for that pixel. For each receptive field, color scale is normalized to the maximum weight and centered around 0. Right: preferred angle is max of marginal. (D) Top: anatomical location of tectal neuropil major axes (left: green, right: purple). Middle and Bottom: Average preferred angle gradient along left and right axis, shaded area is standard deviation. Middle: Prey-experienced (N = 23, blue). Bottom: Prey-naïve (N = 19, red). (E) Average correlation values of visual area pixels in pretectum (left), tectal neuropil (middle), and tectal PVNs (right) were not significantly different between prey-experienced (N = 23) and prey-naïve fish (N = 17), p = 0.80 for pretectum; p = 0.42. (F) Average distribution of pixels' preferred angles in each area (columns) in prey-experienced (blue, top row) and prey-naïve (red, bottom row) fish. There were no differences in average preferred angle distributions between the two groups of fish (two-sample Kolmogorov-Smirnov tests, p = 0.93 for pretectum, p = 0.94 for tectal neuropil and p = 0.95 for tectal PVNs.). See also *Figure 3—figure supplement 1*. A permutation test was used for all pairwise comparisons if not specified otherwise (see Materials and methods, Behavioral data analysis and statistics). Data tables for panels B, C, D, E, and F in *Figure 3—source data 1*.

The online version of this article includes the following source data and figure supplement(s) for figure 3:

**Source data 1.**
**Figure supplement 1.** Similar retinotopic maps and pretectal events in prey-experienced and prey-naïve fish.
**Figure supplement 1—source data 1.**

To validate the use of Granger-causality analysis for calcium imaging in prey capture, we first applied it within the visual system where the basic circuitry is well characterized (*Scott, 2009*; *Nevin et al., 2010*; *Preuss et al., 2014*). We analyzed signals across the entire recording period. We compared baseline activity to activity evoked in the presence of a paramecium in both prey-experienced and prey-naïve fish (*Figure 4B*). We found that the tectal neuropil (*Figure 4B and C*, areas 3 and 4) predicts activity in tectal PVNs (*Figure 4B and C*, areas 5 and 6). This is consistent with neuro-anatomical connections between the two regions in which PVN dendrites in the neuropil receive input from upstream tectal neurons (*Figure 4D*; *Scott, 2009*; *Nevin et al., 2010*; *Preuss et al., 2014*). Similarly, the information flow from the pretectum to tectal PVNs could be explained by pretectal neurons projecting to superficial layers of the optic tectum where PVN dendrites ramify (*Figure 4D*; *Semmelhack et al., 2014*; *Scott, 2009*; *Nevin et al., 2010*). Thus, statistical causality analysis of calcium activity imaging data during prey capture agrees with known functional and anatomical connections in visual areas.

In the presence of the prey, the strength of information coupling between visual regions increased relative to baseline spontaneous activity (*Figure 4B,C*). Specifically, we observed stronger statistical interactions between the pretectum and the OT, and between left and right hemispheres of the OT, consistent with the central role of binocular vision in prey capture (*Figure 4C*). The anatomical basis for this functional connection needs further investigation. Interestingly, prey-experienced fish did not differ statistically from prey-naïve fish in connectivity strength in visual areas during either spontaneous activity or in presence of prey (*Figure 4B*). This provides additional evidence suggesting experience of prey does not affect information flow between visual areas.

## Connectivity between visual areas and other areas of the prey capture circuit

To determine if communication between brain areas is altered by experience of live prey, we applied Granger-causality analysis to understand region-to-region dynamic interactions and potential differences between prey-experienced and prey-naïve fish (*Figure 5A–B*). Consistent with what is known about physical connectivity (*Scott, 2009*; *Nevin et al., 2010*; *Bae et al., 2009*), activity in the tectal neuropil predicted activity in the tectal PVNs, activity in the PVNs predicted activity in the cerebellum, and activity in the cerebellum predicted activity in the hindbrain in both prey-experienced and prey-naïve fish. Other significant interactions observed in both prey-experienced and prey-naïve fish included from tectal PVNs to tectal neuropil (consistent with *Nevin et al., 2010*), from cerebellum to tectal PVNs (as suggested anatomically by cerebellar output neurons that project to the optic tectum *Heap et al., 2013*), and from hindbrain to tectal neuropil (as shown in the *Xenopus* tadpole *Hiramoto and Cline, 2009*). When comparing prey-experienced and prey-naïve fish, we found no statistical difference in the apparent functional connectivity between visual and motor-related areas,

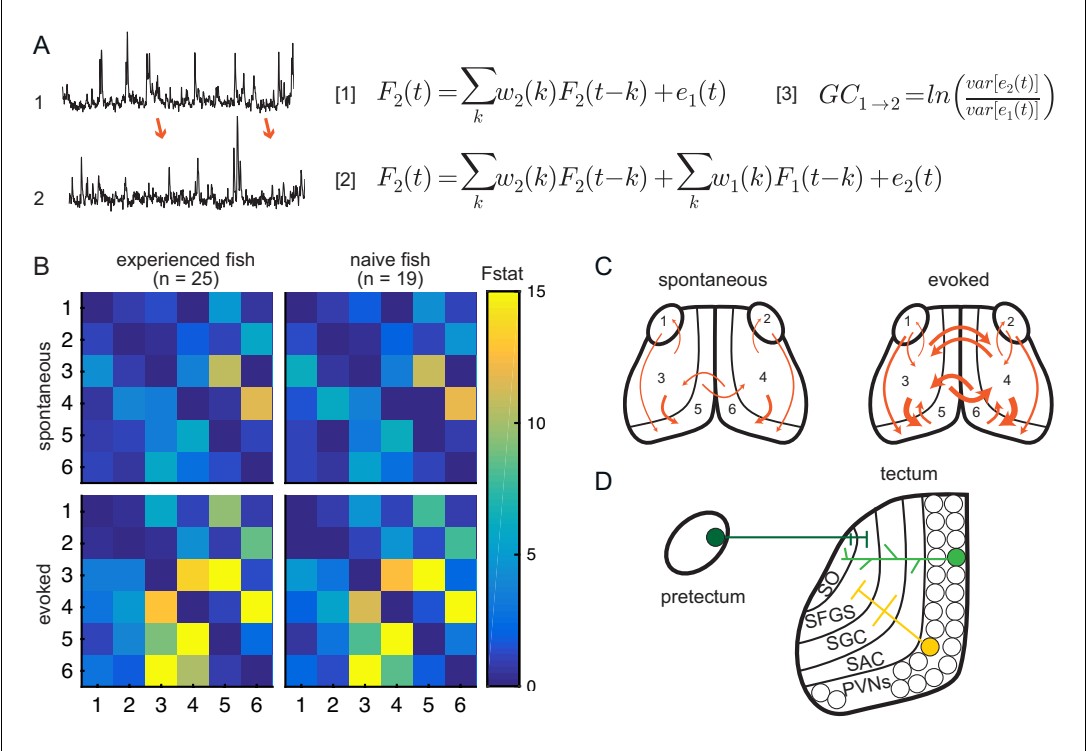

**Figure 4.** Experience does not affect directed information flow between visual areas. (A) Granger-causality equations (right) to model fluorescence time-series 2 (TS2) using information from TS1 (left). TS1 and TS2: fluorescence from region 4 and 3 respectively for one representative fish. F(t) is fluorescence for time point t; $w_1$ and $w_2$ are the weights calculated for each time point; e denotes prediction error. *Figure 4 - equation 1* $F_2(t) = \sum_k w_2(k)F_2(t-k) + e_1(t)$ and *Figure 4 - equation 2* $F_2(t) = \sum_k w_2(k)F_2(t-k) + \sum_k w_1(k)F_1(t-k) + e_2(t)$ are the autoregressive models for univariate and bivariate signals, respectively. *Figure 4 - equation 3* is estimation of Granger-causality level. (B) Average causality level within visual areas in spontaneous (no prey, top row) and evoked (prey present, bottom row) conditions, in prey-experienced (left column) and prey-naïve (right column) fish. Each box represents the F-statistic which quantifies the statistical significance of the directed interaction from the region identified by the row to the region identified by the column. F-statistic values ranged from 0 to 23.7. Fstat values above 15 are yellow. Brain areas shown are: 1, left pretectum; 2, right pretectum; 3, left tectal neuropil; 4, right tectal neuropil; 5, left tectal PVN; 6, right tectal PVN. Significant causal interaction causality link for Fstat > 3.88. No significant difference between prey-experienced and prey-naïve fish in either spontaneous or evoked Granger-causality matrices (pairwise ts, corrected using the Benjamini-Hochberg False Discovery Rate (FDR), see Materials and methods, Behavioral data analysis and statistics; see *Supplementary file 2* for p-values). (C) Schematics of functional links in visual areas in spontaneous (left) and evoked (right) conditions. Line width proportional to Granger-causality level (evoked and spontaneous maps indicate links with Fstat > 3.88). (D) Anatomy and known connections of the optic tectum. Dark green: input from pretectum to OT. Bright green: PVNs with dendritic arborization in tectal neuropil. Yellow: axonal projections from PVNs to different layers of OT. SO, *stratum opticum*; SFGS, *stratum fibrosum et griseum superficiale*; SGC, *stratum griseum centrale*; SAC, *stratum album centrale*. See also *Figure 4—figure supplement 1*. Data table for panel B in *Figure 4—source data 1*.
The online version of this article includes the following source data and figure supplement(s) for figure 4:

**Source data 1.**
**Figure supplement 1.** Experience does not affect circuit covariance in visual areas.
**Figure supplement 1—source data 1.**

however the functional link from tectal PVNs to telencephalon was significantly greater in prey-experienced fish (*Figure 5A* and *Supplementary file 3*).

To better understand how the apparent region-to-region information flow relates to prey capture initiation, we compared fish that initiated prey capture often ('strong' hunters) *versus* rarely ('weak' hunters). Among prey-experienced fish, strong hunters showed enhanced information flow from pretectum to the tectal neuropil, tectal PVNs, cerebellum and hindbrain, but not to the telencephalon or habenula (*Figure 5—figure supplement 1A*). Together, these results suggest first, an important role of the pretectum and its connectivity to downstream visual and motor-related areas in determining frequency of prey capture initiation, and second, a possible implication of information flow from

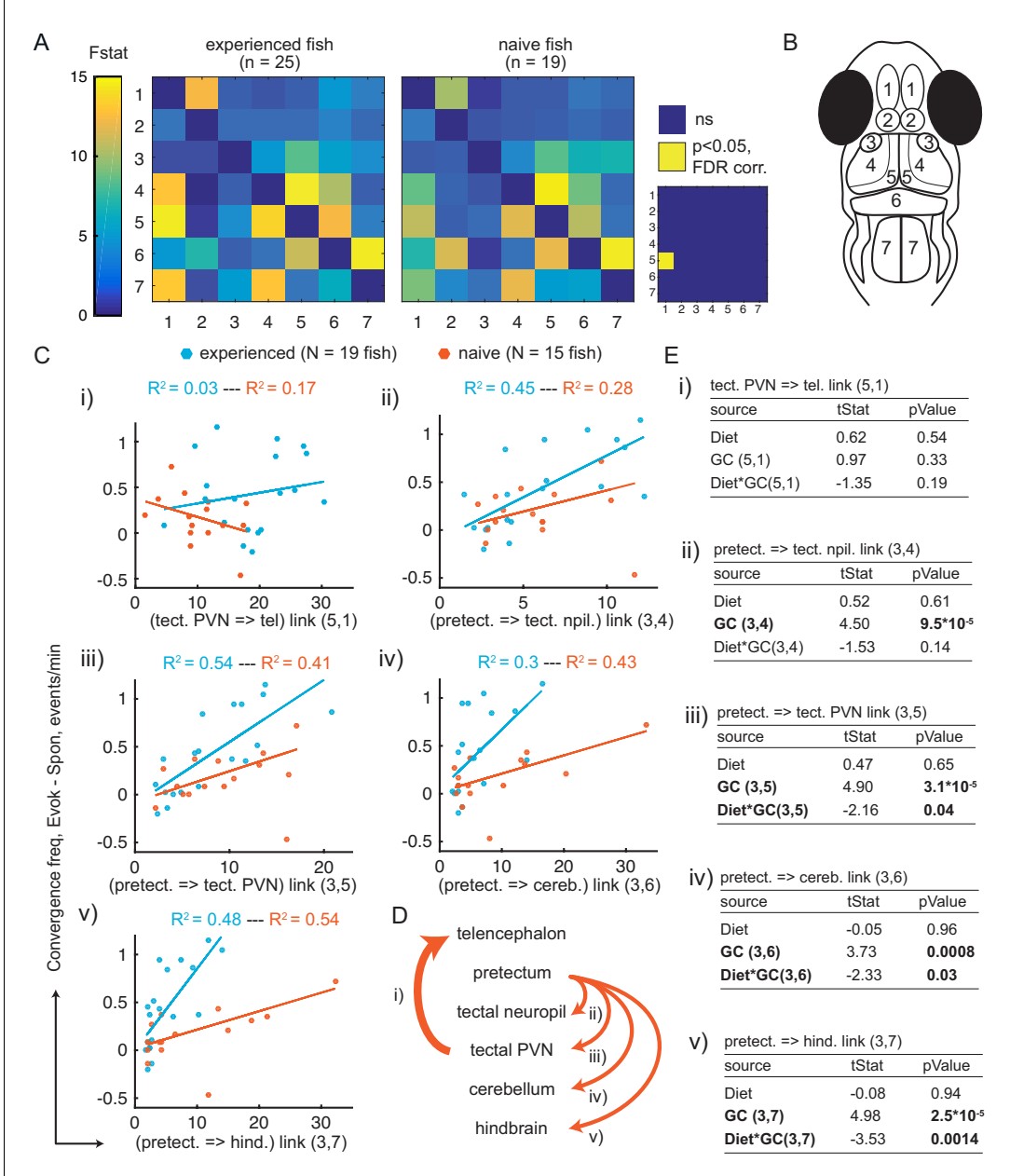

**Figure 5.** Granger-causality-based estimation of interactions between visual and motor areas correlates with prey capture initiation. (A) Average Granger-causality between brain areas in prey-experienced (left) and prey-naïve (right) fish. Activity from left and right sides averaged as depicted in anatomical schematic (B). F-statistics ranged from 0 to 34.1. Inset: Pairwise statistical comparison of all links. Significant interactions represented in yellow (p < 0.05, pairwise Ts, FDR corrected, see Materials and methods, Behavioral data analysis and statistics; see *Supplementary file 3* for p-values). (B) Brain areas shown in schematic are: 1: telencephalon, 2: habenula, 3: pretectum, 4: tectal neuropil, 5: tectal PVNs, 6: cerebellum, 7: hindbrain. (C–E) Granger-causality statistic is significantly correlated (p-values in E, 'GC' row in each table) with eye convergence frequency for interactions from pretectum to downstream areas (interactions shown to be significantly stronger in 'strong' hunters, see *Figure 5—figure supplement 1*), but not for interaction from tectal PVNs to telencephalon (interaction shown to be significantly stronger in prey-experienced fish in A). Interaction between experience of prey ('Diet') and Granger-causality strength was significant for pretectum to tectal PVN, pretectum to cerebellum, and pretectum to hindbrain (p-values in E, 'Diet*GC' row in each table). (C) Eye convergence frequency (evoked – spontaneous) as a function of Granger-causality strength for: (i) tectal neuropil→telencephalon (5,1), (ii) pretectum→tectal neuropil (3,4), (iii) pretectum→tectal PVN (3,5), (iv) pretectum→cerebellum (3,6), (v) pretectum→hindbrain (3,7). Statistics of linear regression model are in (E). (D) Schematic of links considered in (C). (E) Robust linear regression model: [Convergence Frequency ~1 + Diet + GC + Diet*GC], where 'Convergence Frequency' is (with prey – without prey), 'GC' is Granger-causality Fstat, 'Diet' is prey-experienced or prey-naïve fish (categorical variable), and 'Diet*GC' is interaction between experience of prey and Granger-causality

*Figure 5 continued on next page*

*Figure 5 continued*

statistic. N = 19 and N = 15 prey-experienced and prey-naïve fish respectively. Significant terms are bolded, GC for all links but link (5,1), and (GC*diet) interactions for links (3,5 , 3,6) and (3,7). Data table for panel A and C in *Figure 5—source data 1*.

The online version of this article includes the following source data and figure supplement(s) for figure 5:

**Source data 1.**
**Figure supplement 1.** Comparison of weak vs strong hunters.
**Figure supplement 1—source data 1.**

tectal PVN to telencephalon in mediating the effect of experience of live prey on subsequent initiation frequency.

## Experience of live prey increases the probability of transitioning from sight of prey to capture initiation

Having observed an augmented drive from tectal PVNs to telencephalon in prey-experienced fish (*Figure 5A*), and from pretectum to downstream brain areas in strong hunters (*Figure 5—figure supplement 1*), we asked how experience of prey affects the relationship between these connectivity patterns and prey capture initiation frequency. For both prey-experienced and prey-naïve fish, we found a linear relationship between eye convergence frequency and dynamical drive from pretectum to tectal neuropil, tectal PVNs, cerebellum and hindbrain (*Figure 5C–E*). Statistical causality strength was correlated with variance in behavior ($0.28 < R^2 < 0.54$, *Figure 5C* ii-v). However, there was no significant relationship with interaction strength from tectal PVNs to telencephalon (*Figure 5Ci–Ei*). Comparing prey-experienced and prey-naïve fish, we found a significant interaction between experience of live prey and statistical causal strength in predicting prey capture initiation for links from pretectum to cerebellum and hindbrain (3- to 4-fold increase in slope for prey-experienced fish), and tectal PVNs (2-fold increase in slope) (*Figure 5C–E*). For example, for a given level of pretectum drive to cerebellum, eye convergence frequency was higher in prey-experienced fish *versus* prey-naïve fish, suggesting the system is sensitized and more likely to trigger a capture. These results suggest that experience of live prey increases likelihood of triggering capture initiation for a given level of information flow from pretectum to downstream areas.

Our observations of prey-evoked activity suggest the forebrain may play a role in prey capture since the telencephalon and habenula are activated during eye convergence (*Figure 2H*) and the directed interaction from tectal PVNs to telencephalon is significantly stronger in prey-experienced fish (*Figure 5A–B*). To probe involvement of the forebrain specifically in prey capture initiation, we compared pretectal transients associated with eye convergence (i.e., 'fish has detected prey and initiates capture') to ones that do not lead to motor output (i.e., 'fish has detected prey but does not initiate capture') (see Materials and methods, Calcium and behavior imaging data pre-processing for detection of pretectal transients). As expected, in both prey-experienced and prey-naïve fish, activity in the cerebellum and hindbrain was only detected when there was a tail flick and eye convergence (*Figure 6A–C*). In contrast, pretectal activation was high in both eye convergence and pretectum-only events, although it lasted longer during eye convergence events (*Figure 6A*). Patterns of activity in visual areas downstream of the pretectum were similar in prey-experienced and prey-naïve fish in both states, however the probability of a pretectal event being followed by an eye convergence was significantly larger in prey-experienced fish (*Figure 6D*).

The only brain region where we detected a marked difference between prey-experienced and prey-naïve fish was the forebrain (*Figure 6A–B*). We compared activity in the telencephalon and habenula between prey-experienced and prey-naïve fish and found a significant difference during pretectal transients associated with eye convergence relative to pretectal transients not accompanied by behavior (*Figure 6E*).

These observations suggest that similar pretectal events can either remain confined in the visual areas or activate motor areas thereby triggering eye convergence. The telencephalon and/or habenula may operate as a switch that, when activated favors initiation of prey capture in response to prey-evoked visual activity.

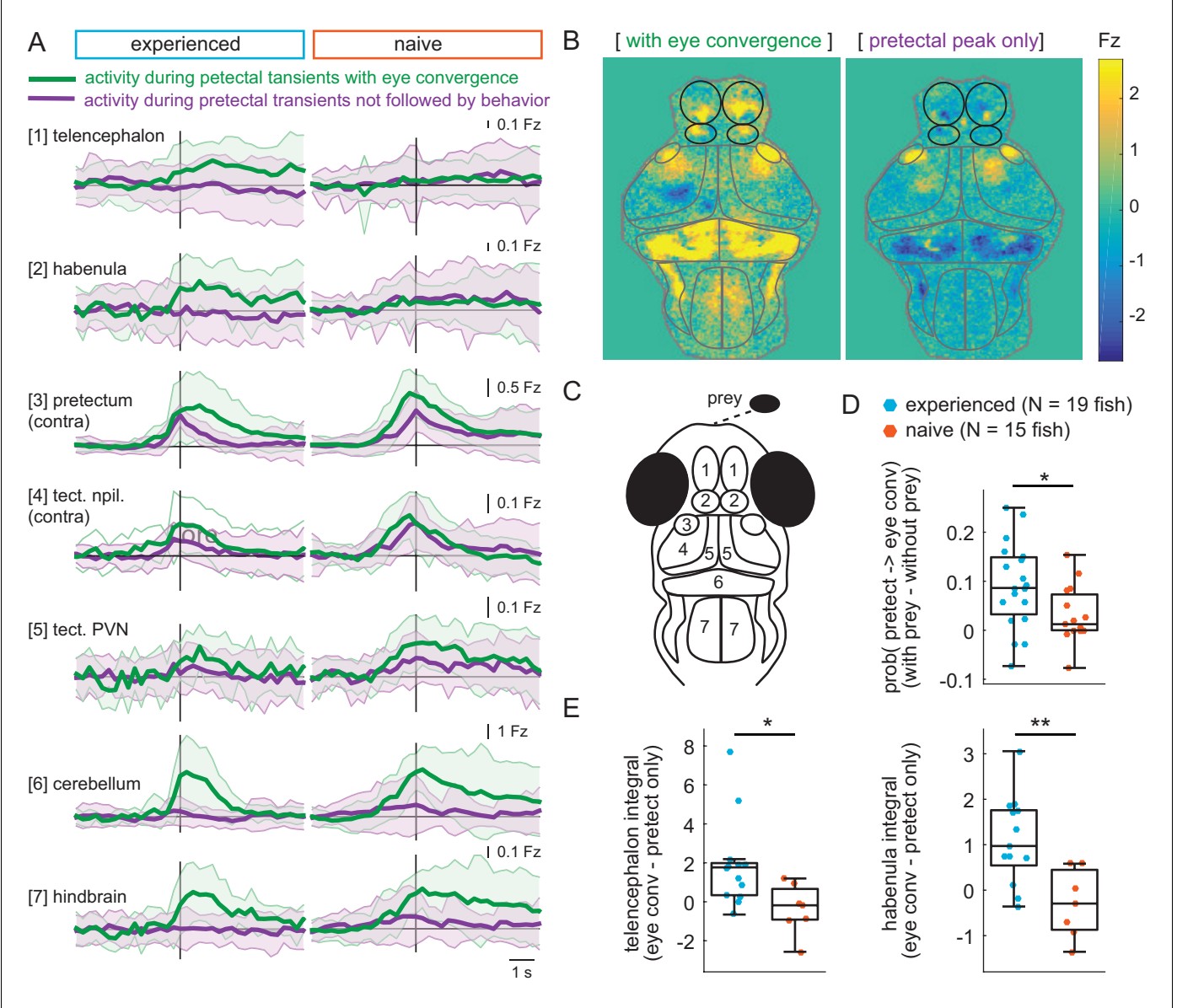

**Figure 6.** Experience of prey increases capture initiation-associated forebrain activity and lowers threshold for visual activity to trigger capture initiation. (A) Time traces of activity in seven brain regions during either a pretectum transient followed by eye convergence (green) or a pretectum transient with no behavior (purple) in a representative prey-experienced fish (left, N = 12 convergences, N = 42 pretectum-only events) and prey-naïve fish (right, N = 15 convergences, N = 20 pretectum-only events). Black vertical line represents the pretectal activity peak to which transients are aligned. 'Contra' and 'ipsi' refer to the side with higher or lower pretectal transient amplitude peak time (see Materials and methods). (B) Average brain activity maps for another representative prey-experienced fish in presence of prey, showing summed calcium activity over 4.2 s (five frames before to 10 frames after event), during either pretectum transients associated with eye convergence (prey to left or right of the fish, N = 22) or pretectum transients not accompanied by behavior (N = 34). Brain areas are outlined in gray as in schematic C, and forebrain areas are additionally outlined in black. (A and B) show that forebrain areas appear active during pretectal transients associated with eye convergence, but not pretectal transients not accompanied by behavior. (C) Schematic of anatomical areas considered in A and B. (D) Prey-experienced fish (N = 19) have a higher probability of pretectum transients being associated with eye convergence than prey-naïve fish (N = 15), suggesting visual events are more likely to cause motor output with experience of live prey (* indicates p = 0.03). Raw values of evoked and spontaneous probabilities are not significantly different in experienced *versus* naïve fish (reported in *Figure 6—source data 1*). (E) Prey-experienced fish (blue) have significantly more telencephalon (left, p = 0.02) and habenula (right, p = 0.004) activity than prey-naïve fish (red) during pretectal transients associated with eye convergence relative to pretectal transients not accompanied by behavior. Box plot shows difference in fluorescence integral five frames before to five frames after events (pretectal transient with vs. without eye convergence). Fish with < 5 eye convergences were excluded, prey-experienced fish N = 13, prey-naïve fish N = 7. * and ** indicate p < 0.05 and p < 0.01 respectively. A permutation test was used for all pairwise comparisons if not specified otherwise (see Materials and methods, Behavioral data analysis and statistics). Data tables for panels A, D, E, and F are in *Figure 6—source data 1*.

*Figure 6 continued on next page*

*Figure 6 continued*

The online version of this article includes the following source data for figure 6:

**Source data 1.**

## Forebrain disruption reduces hunting initiation in prey-experienced fish

Our results so far suggest that, in prey-experienced animals, the forebrain is recruited during prey-elicited activity in a visual area and that this forebrain activity increases the probability of activation of motor areas and, thus, prey capture behavior in experienced fish. If correct, this model predicts that disruption of forebrain activity should compromise prey capture behavior. We sought to test this prediction by chemical ablation of cells in the forebrain. To do this, we expressed the gene encoding the enzyme nitroreductase (NTR) in transgenic *Tg(gng8:Gal4;UAS:NTR-mCherry)* larvae (*Figure 7A*). The *Tg(gng8:Gal4)* line drives expression in the dorsal habenula and its projections to the interpeduncular nucleus, with a small amount of labeling of mitral cells in the olfactory bulb (*deCarvalho et al., 2013*; *Figure 7A* and *Figure 7—figure supplement 1C*). Since prey capture is a visually-guided behavior (*Johnson, 2019*), which does not depend on olfactory cues (*Patterson et al., 2013*; *Muto et al., 2017*), we conjectured that any effect of the manipulation on prey capture would be due to disruption of the habenula and not due to an effect on olfaction. NTR transforms the innocuous antibiotic metronidazole (MTZ) into a toxic metabolite, resulting in death of expressing cells (*Pisharath et al., 2007*; *Figure 7B–C* and *Figure 7—figure supplement 2*). Prey-experienced larvae not expressing NTR showed no significant difference in number of eye convergences between MTZ-treated and control animals (*Figure 7D*). In contrast, fish expressing NTR showed a significant decrease in eye convergence frequency when treated with MTZ in a free-swimming prey capture assay (*Figure 7E*) and spent a significantly lower percentage of time hunting with eyes converged (*Figure 7F*). In accordance with this effect, MTZ had no effect on paramecia consumption in control NTR- fish (*Figure 7—figure supplement 1A*), while MTZ treatment significantly reduced the number of paramecia consumed in the recorded period for NTR+ fish compared to siblings treated only with the DMSO vehicle (*Figure 7—figure supplement 1B*). Swimming behavior during the acclimation and prey capture periods of the hunting assay was not affected by MTZ (*Figure 7—figure supplement 1E and F*). Moreover, MTZ had no effect on paramecia consumption on day 7 dpf in naïve fish, which were fed with flakes on day 5 and 6 dpf (*Figure 7—figure supplement 1G*). Together, these findings suggest that activity in the habenula is specifically required for enhanced prey capture performance in prey-experienced fish.

## Discussion

### Experience of live prey improves hunting success in larval zebrafish

After larval zebrafish hatch from their chorion and finish using the nutrient reserves from their yolk, they are left to their own devices to survive, avoid predators, and capture prey. Prey capture is generally thought to be an innate behavior in zebrafish because larvae are capable of capturing prey as early as their first attempts (*Borla et al., 2002*; *McElligott and O'Malley, 2005*; *McClenahan et al., 2012*). We compared hunting of paramecia between zebrafish larvae with two days of experience of live prey and sibling fish with experience of inert dry food, which are exposed to paramecia for the first time. We found that prior experience of live prey increases the frequency of successful captures, indicating that 'practice' improves performance of this innate behavior. This improvement can be measured both in freely swimming fish as well as when fish are semi-immobilized for imaging and observing a live prey. Moreover, we find that experience of paramecia generalizes, increasing responsiveness to a virtual prey, even though this small black dot moving at uniform speed on a screen is quite distinct from the erratic 3D movement patterns of a translucent paramecium. While dependence of virtual prey capture on experience of live prey has not been explicitly described previously, we note that studies on prey capture in a virtual environment consistently report feeding the fish paramecia prior to testing (*Bianco et al., 2011*; *Trivedi and Bollmann, 2013*; *Semmelhack et al., 2014*; *Bianco and Engert, 2015*; *Henriques et al., 2019*). Thus, experience of live prey may contribute to the ontogeny of prey capture behavior, where older (and therefore more

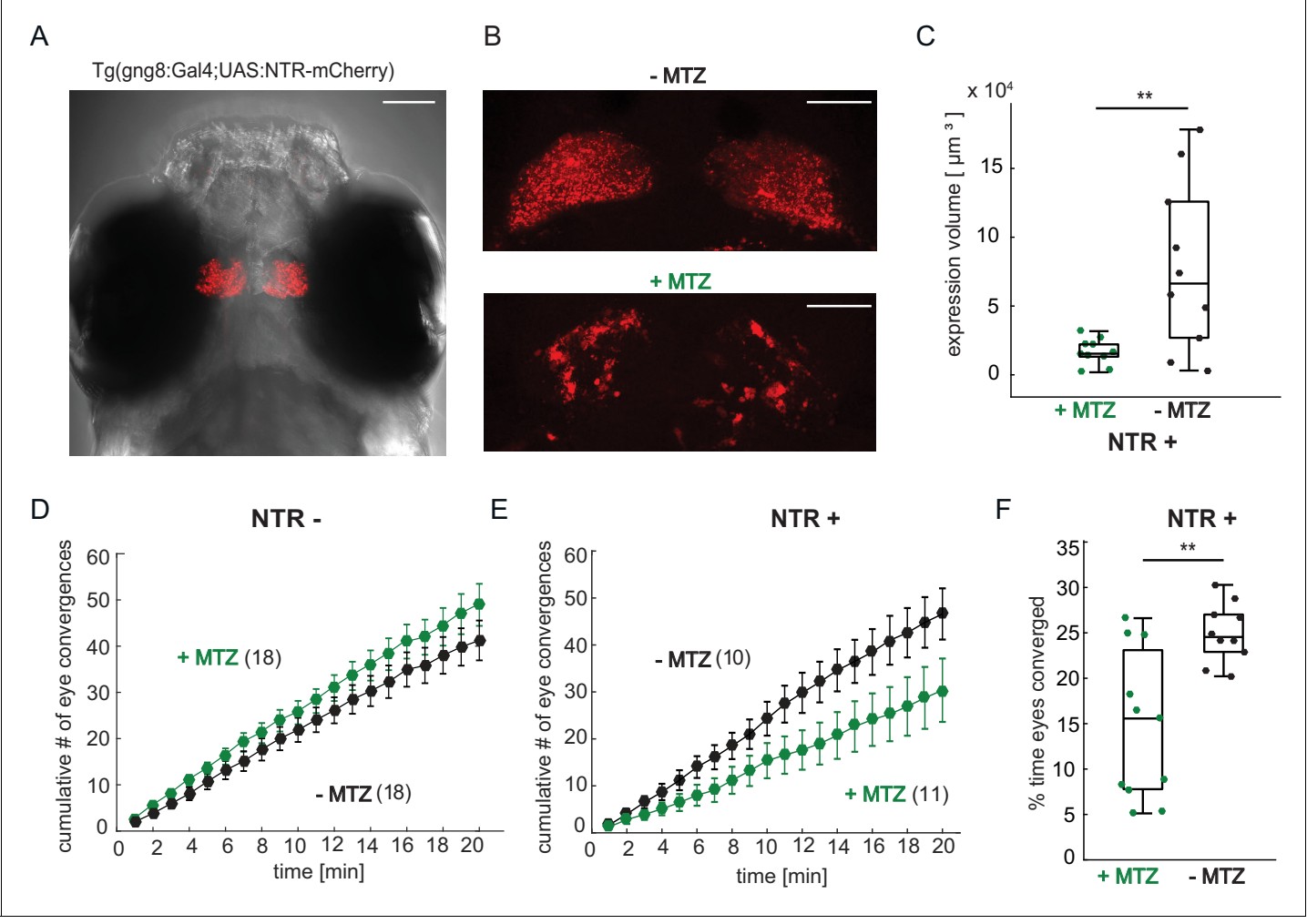

**Figure 7.** Chemical disruption of the habenula reduces hunting behavior in prey-experienced fish. (A) Representative image of 7 dpf *Tg(gng8:Gal4; UAS:NTR-mCherry)* fish, dorsal view. NTR-mCherry is depicted in red, image is a maximum intensity projection of a Z-stack. Additional example image is shown in *Figure 7—figure supplement 1C*). Scale bar is 100 μm. (B) Representative images of NTR-mCherry expression (red) in the habenula in NTR+ fish after 19 hr in either 0.2% DMSO (-MTZ) (top) or 5 mM MTZ in 0.2% DMSO (+MTZ) (bottom). Images are maximum intensity projections of 5 μm/slice Z-stacks. Scale bar is 40 μm. Additional example images are shown in *Figure 7—figure supplement 2A*. (C) Volume of NTR-mCherry fluorescence is reduced in NTR-mCherry expressing fish treated with MTZ, p = 0.003, N = 10 animals per group. Differences in signal likely reflect variable transgene expression. Dependence of volume measurement on total fluorescence differs between +MTZ and -MTZ (*Figure 7—figure supplement 2B*). Symbols indicate individual fish. Box plot shows median, 25th and 75th percentiles. (D, E) Cumulative average number of eye convergences after addition of paramecia in 7 dpf control (NTR-) fish (D) and *Tg(gng8:Gal4;UAS:NTR-mCherry)* (NTR+) siblings (E), pretreated for 19 hours with either 0.2% DMSO alone (-MTZ) or 0.2% DMSO containing 5 mM metronidazole (+MTZ) and tested for 20 minutes one hour after washout of the drug. Eye convergence rate is reduced in MTZ-treated enzyme expressing fish (NTR+/+MTZ) compared to untreated (NTR+/-MTZ) animals (two-way repeated measures ANOVA shows an effect of treatment with p = 0.033), while we observe no significant difference for NTR- animals (two-way repeated measures ANOVA, effect of treatment p = 0.24). (n) = number of fish in each group. Error bars show SEM. (F) Percent time of eyes converged over 20 min of recording period in NTR+ fish is significantly lower in NTR+/+MTZ fish than in NTR+/-MTZ (p = 0.004. Symbols indicate individual fish. Box plot shows median and 25th and 75th percentiles. A permutation test is used for all pairwise comparisons unless otherwise specified (see Materials and methods, Behavioral data analysis and statistics). Data tables for panels C, D, E, and F in *Figure 7—source data 1*.

The online version of this article includes the following source data and figure supplement(s) for figure 7:

**Source data 1.**

**Figure supplement 1.** Expression pattern of *Tg(gng8:Gal4;UAS:NTR-mCherry)* fish, paramecia consumption and swim behavior.

**Figure supplement 1—source data 1.**

**Figure supplement 2.** Differences in mCherry signal intensity and correlation with expression volume.

**Figure supplement 2—source data 1.**

'experienced') fish have more fluid capture maneuvers and hunt prey from a wider angular range (*Westphal and O'Malley, 2013*). Our finding is consistent with a recent study (*Lagogiannis et al., 2019*), which showed that differences in hunting kinematics result in higher hunting efficiency in experienced fish.

Genetically encoded circuits extract information relevant to predation at multiple levels of the visual processing stream. Different groups of retinal ganglion cells and optic tectum neurons respond preferentially to small prey-like objects, or looming predator-like objects in zebrafish (*Semmelhack et al., 2014*; *Del Bene et al., 2010*; *Preuss et al., 2014*; *Temizer et al., 2015*), and frogs (*Ewert et al., 2001*), similarly to small target motion detector neurons in insects (*Barnett et al., 2007*; *Wiederman et al., 2013*). This information is relayed in the tectum, whose optogenetic activation has been shown to trigger the prey capture motor response (*Fajardo et al., 2013*). Further, recent results have shown that stimulation of single neurons in the pretectum can trigger the hunting sequence (*Antinucci et al., 2019*). Thus, the prey capture circuit is hard-wired in the novice hunter's brain. Nonetheless, experience of live prey may shape the circuit as shown in this work and the work mentioned above (*Lagogiannis et al., 2019*), and as it appears to do in juvenile fish raised in the dark who learn to forage using their lateral line system (*Carrillo and McHenry, 2016*).

## Activation of visual areas during prey observation and capture initiation

Our behavioral analysis showed an increase in prey capture initiation and successful captures in prey-experienced fish. This observation led us to test whether experience of live prey (vs. experience of dry inert food) enhances the early step of prey visualization or a later step of 'decision to pursue'. To distinguish between these possibilities, we imaged neural activity in semi-immobilized prey-experienced and prey-naïve fish in response to a live prey. Regression analysis of calcium signals revealed retinotopic maps that encode prey position in the pretectum and optic tectum. Previous findings show that spontaneous activity in the optic tectum changes with development and visual experience (*Avitan et al., 2017*). We observed no difference in maps between prey-experienced and prey-naïve fish, suggesting that neural encoding of prior experience of prey lies in a step subsequent to visualization of prey.

We examined brain activity around eye convergences, the hallmark of prey capture initiation (*Bianco et al., 2011*). In agreement with an earlier finding that AF7 responds specifically to prey-like objects and is important for mediating the capture response (*Semmelhack et al., 2014*), we observed that activation of pretectal neurons, likely surrounding AF7, consistently preceded eye convergence (*Figure 2*). We also found that activation of the tectal neuropil and PVNs preceded eye convergence, consistent with previous observations showing that optic tectum function is necessary for prey capture (*Gahtan, 2005*; *Smear et al., 2007*; *Del Bene et al., 2010*), and that assemblies of tectal neurons activate specifically when an eye convergence occurs (*Bianco and Engert, 2015*). The amplitude and timing of activity observed in visual areas in the presence of prey was similar in prey-experienced and prey-naïve fish whether it elicited an eye convergence or not (*Figures 3* and *4*, and *Figure 3—figure supplement 1*). Thus, neither the mapping of prey-evoked activity in visual areas nor strength or timing of visual responses appeared to be altered by experience of live prey (vs. experience of dry inert food). We therefore turned to an analysis of motor-related areas inducing prey capture initiation.

## Circuit activation during prey capture initiation

When fish initiate a prey capture sequence, they converge their eyes and flick their tail in a characteristic J-bend. Consistent with this, prey capture initiation events were associated with activation of the cerebellum and hindbrain (*Figure 2*). Similar to mammals, the teleost cerebellum is compartmentalized into the vestibulo-cerebellar and the non-vestibulo-cerebellar systems that control balance and locomotion respectively (*Volkmann et al., 2008*), which likely play a role in the outcome of a capture sequence. To understand how information flows from visual areas to these motor-related areas, we used Granger-causality analysis (*Seth et al., 2015*). We established that the strength of apparent directed information flow from pretectum to optic tectum, cerebellum, and hindbrain were significantly correlated with how frequently a fish initiated a hunting sequence (*Figure 5C–E*). This relationship could reflect a role in triggering the motor response to the sight of prey or a change in

proprioceptive feedback due to movement. A reason for favoring the former interpretation is that proprioception would be expected to be the same in prey-experienced and prey-naïve fish during an eye convergence event. Instead we observe a difference in slope of directed coupling strength relative to eye convergence frequency for information flow from pretectum to tectal PVN, from pretectum to cerebellum, and from pretectum to hindbrain (*Figure 5C–E*). For a statistical link with a given strength from the pretectum to motor-related areas, fish with prior experience of live prey initiate prey captures more frequently than those with prior experience of dry inert food. Our results suggest that prior experience of prey increases the gain of information flow from the pretectum to motor-related areas, making a pretectum event more likely to trigger a prey capture response.

## Forebrain recruitment and the experience-dependent boost in prey capture initiation

Wide-field brain activity imaging showed that prey-experienced fish had greater activity in the telencephalon and habenula when visual activity is followed by eye convergence relative to their prey-naïve siblings (*Figure 6*). Statistical causality analysis suggested that prior experience of prey increases information flow from tectal PVNs to the telencephalon (*Figure 5*), suggesting a role for this connection in improved prey capture performance (*Figure 1*). Together, these results suggest forebrain regions activated by the visual system's response to prey may play a role in setting the gain of information flow through the prey capture circuit, with prior experience of prey sharpening the contrast between visual events that do or do not trigger capture behavior.

Since the above observations were correlative, we endeavored to perform a causal test of the role of the forebrain in prey capture. To do this, we turned to targeted chemical ablation using nitroreductase in combination with metronidazole. We targeted the habenula for three reasons: availability of a habenula-specific Gal4 driver line, the fact that the habenula receives considerable anatomical input from the ventral telencephalon (subpallium) (*Turner et al., 2016*), and our observation from Granger-causality analysis that activity in the habenula is predicted by activity in the telencephalon (*Figure 5A*). We selectively disrupted neurons in the habenula following two days of exposure to live prey and observed a significant reduction in prey capture initiation and performance in fish with disrupted habenula neurons vs. siblings with an intact habenula. Our results indicate that activity in the habenula contributes to optimized prey capture performance in prey-experienced fish by increasing the gain between prey-evoked activity in visual areas and the motor areas that execute hunt behavior.

To our knowledge, while there are correlative studies for the role of the telencephalon and habenula in prey capture, no functional role for these structures had yet been demonstrated (*Ewert et al., 1999*). The dorsal habenula in teleosts is homologous to the mammalian medial habenula and has been shown to modulate fear responses and social conflict outcomes in adult zebrafish (*Agetsuma et al., 2010*; *Chou et al., 2016*), and integrating olfactory and optical cues in larval zebrafish (*Jetti et al., 2014*). Our work suggests the forebrain could mediate the effect of prior experience of live prey on prey capture performance. Additional work is required to detail the mechanism of this effect. It should be noted that other brain areas may also contribute to gain control. Previous studies identified the nucleus of the medial longitudinal fasciculus (nMLF) as an important relay for motor signals controlling the prey capture circuit (*Gahtan, 2005*), which was missing from our imaging plane. Some pretectal neurons have direct projections to the nMLF and the hindbrain (*Semmelhack et al., 2014*; *Antinucci et al., 2019*; *Helmbrecht et al., 2018*). The mesencephalic reticular formation, a region controlling eye movements and convergence in goldfish (*Angeles Luque et al., 2005*) was also missing from our imaging plane. Neuromodulation could also contribute to the gain control of prey capture behavior observed in this study. The dopaminergic system encodes stimulus valence and regulates motivation *Matsumoto and Hikosaka, 2009*; the noradrenergic locus coeruleus is thought to modulate arousal *Carter et al., 2010*; different parts of the hypothalamus are thought to control a variety of motivational functions like arousal and feeding (*Mahler et al., 2014*; *Wee et al., 2019*). The central amygdala has recently been shown to control predatory hunting in mice, by increasing capture initiation via the periaqueductal gray (*Han et al., 2017*), which is homologous to the *griseum centrale* of zebrafish, and receives input from the habenula (*Olson et al., 2017*). Further, the habenula, which is a highly conserved across species, also acts in value-based decision making throughout species (*Hikosaka, 2010*), and this could contribute to the experience-dependent increase in prey capture performance that we find to be associated with

habenula activity. The serotonergic system modulates responsiveness and arousal in fish (*Yokogawa et al., 2012*) and mammals (*Boulougouris and Tsaltas, 2008*), as well as prey-approach behavior depending on hunger levels of the fish (*Filosa et al., 2016*). A recent study showed that a subpopulation of neurons in the dorsal raphe in the hindbrain encodes whether zebrafish are in an hunting 'exploitation' state or an 'exploration' state (*Marques et al., 2020*). This area is another candidate for contributing to the effect of experience of live prey on behavior. Further, another study built a detailed model categorizing zebrafish behaviors, and showed that hunger influenced animals' likelihood to seek food vs. safety (*Johnson et al., 2020*).

In summary, we show that fish with prior experience of prey are better hunters and respond more reliably to virtual prey. Prey-experienced fish are more likely to trigger a capture initiation in response to a given visual neural event. Prey-experienced fish also display greater activity in the forebrain during visual events that trigger capture behavior relative to those that do not. Finally, prey-experienced fish show strengthened functional links from the output neurons of the optic tectum to the telencephalon. These findings suggest a role for the forebrain in prey capture, and this is supported by the observation that disruption of one of the forebrain areas, the habenula, compromises prey capture in prey-experienced fish. We hypothesize that experience in hunting of live prey boosts prey capture performance by increasing the impact of information transfer from visual to motor-related areas as a result of recruitment of forebrain activity. The forebrain activity may contribute to gating this process, with experience sharpening contrast between visual events to create a 'go' / 'no-go' signal for initiating this complex motor behavior.

# Materials and methods

## Key resources table

| Reagent type (species) or resource | Designation | Source or reference | Identifiers | Additional information |
|---|---|---|---|---|
| fish line, *Danio rerio* | AB wild type zebrafish line | ZIRC | ZDB-GENO-960809–7 | |
| fish line, *D. rerio* | TL wild type zebrafish | ZIRC | ZDB-GENO-990623–2 | |
| fish line, *D. rerio* | TgBAC(gng8: GAL4FF); UAS:GFP | *deCarvalho et al., 2013* | | obtained from Halpern Lab |
| fish line, *D. rerio* | Tg(UAS-E1B: NTR-mCherry) | *Curado et al., 2007*; *Matsuoka et al., 2016* | | obtained from Stainier Lab |
| fish line, *D. rerio* | Tg(neurod1: GCaMP6F) | *Rupprecht et al., 2016* | | obtained from Wyart Lab |
| fish line, *D. rerio* | Tg(atoh7: gap43-RFP) | *Zolessi et al., 2006* | | obtained from Wyart Lab |
| chemical reagent | Metronidazole | Sigma-Aldrich | 1442009 USP | |
| organism, *Paramecium caudatum* | Paramecia | ZIRC | Paramecium starter culture | |
| reagent | Fish flakes | Hikari USA | First Bites Specialty fish food | |

## Zebrafish care and transgenic lines

Animal experiments were done under oversight by the University of California Berkeley institutional review board (Animal Care and Use Committee). Adult AB and Tüpfel long fin (TL) strains of *Danio rerio* were maintained and raised on a 14/10 hr light cycle and water was maintained at 28.5°C, conductivity at 500 μs and pH at 7.4. Embryos were raised in blue water (3 g of Instant Ocean salts and 0.2 mL of methylene blue at 1% in 10 L of osmosed water) at 28.5°C. For imaging experiments, fish were screened for GCaMP6f expression at 2 or 3 dpf. We focused our study on early larval stages (5

to 8 dpf) when the neural circuitry of prey detection has been well studied in visual areas, and when animals are more tractable for neural activity imaging due to their transparency and small brain size. We used the *Tg(NeuroD:GCaMP6f)*[icm05] line for all imaging experiments (*Rupprecht et al., 2016*), and the *Tg(atoh7:GAP-RFP)* line (*Zolessi et al., 2006*) was used to compare labeling in the NeuroD line with retinal ganglion cell projections (*Figure 2—figure supplement 1A*). Offspring of a *Tg (gng8:Gal4;UAS:GFP)* (*deCarvalho et al., 2013*) crossed to a UAS:NTR-mCherry fish (*Curado et al., 2007*) were used for the habenula ablation experiments.

## Diet and freely swimming behavior assay in wild type fish

Healthy wild type TL larval zebrafish were selected based on the inflation of the swim bladder at 4 dpf. Fish were split into two groups either fed a diet of paramecia or of fish flakes (Hikari USA inc) with 20 animals per dish. Fresh paramecia were prepared every day. We found that feeding the fish for a minimum of 6 hr per day insured that spontaneous swimming was the same across fish with different diets (*Figure 1—figure supplement 1E and F*). Fish were fed twice a day, in the morning at 9–10 am, and in the afternoon at 1–2 pm. Dishes were cleaned out before each feed and fish were transferred to a new dish every evening at 5–6 pm. Fish were given more food than they could eat to ensure equal levels of satiation (there was always food remaining in the dishes when cleaned). At 7 dpf, one by one, fish were transferred to a 35 mm diameter dish and left to acclimate for one minute under white light. Spontaneous swimming was recorded for five minutes with a uEye CCD camera (IDS Imaging Development Systems GmbH) at 30 Hz using dark field illumination. 500 µL of fresh paramecium culture was then added to the dish and prey capture behavior was recorded for five minutes. There was no significant correlation of initial number of paramecia or time of day the experiment was performed on prey capture performance in either prey-experienced or prey-naïve fish (*Figure 1—figure supplement 1C and D*). We manually counted two types of events for each fish: number of pursuits initiated (eye convergence and J-bend at the same time) and successful captures. Fish that did not move at all during the spontaneous swimming test were excluded. We also compared spontaneous swimming of our two experimental groups to a third group fed pureed brine shrimp and flakes, our fish facility diet (*Figure 1—figure supplement 1E and F*). Finally, to control for differences in brain development, we estimated brain volume using the image analysis software Imaris (Bitplane AG, Switzerland) to interpolate total volume from surfaces drawn manually at 9.22 µm intervals. We found no difference between prey-experienced and prey-naïve fish (data *not shown*, N = 6 fish per group, p = 0.25).

## Virtual prey capture assay

Our study focused on the initiation of prey capture rather than the subsequent motor-sequence. We therefore used an open-loop virtual prey capture assay, as previously described in the literature (*Bianco et al., 2011*; *Trivedi and Bollmann, 2013*). Larval zebrafish that were fed paramecia or flakes were embedded in low-melting point agar at the end of their 6th day. Agar around the eyes and tail was carefully removed so that only the area around the swim bladder was restrained. Fish were kept in the incubator to acclimate overnight. At 7 dpf fish were transferred to our imaging setup, a diffusive filter was fixed to the side of the dish acting as a screen ~10 mm away from the mid-point between the eyes. All stimuli were generated in MATLAB (Mathworks, USA) using the Psychophysics Toolbox extensions (*Brainard, 1997*). All fish were first tested for a robust optokinetic reflex evoked by moving gratings to ensure that the visual system was functional. Fish were then left to acclimate on the setup for 10 minutes. Small moving dots were projected at eye level onto the screen in front of the fish using an M2 Micro Projector (AAXA, USA). Optimal stimulus properties were chosen to maximize prey capture responses: 1 mm diameter dots of varying contrasts on a white background appeared in front of the fish and moved to the left or the right of the screen at 30 degrees/sec. Changes in speed of the stimulus due to the curvature of the screen were corrected for programmatically. The contrast of the dot was varied from 20% (light gray on white) to 100% (black on white) in 20% increments. Dots of different contrasts were presented in blocks. Fish were kept in the dark between trials (12 s inter-trial interval), the white background screen appeared progressively 3 s before the onset of the trial, and at trial onset the stimulus appeared on the screen and moved to the left or to the right for a duration of 3 s. Each contrast was tested eight times (with four in each direction), and 20 blank trials were interweaved randomly with the (8 × 5 contrast types)

target trials throughout the experiment, a total of 60 trials per fish. Contrast blocks were also ordered randomly. Fish were illuminated from the side with a custom-built red LED light source and behavior was imaged with a 2.5x/0.06 air objective (Carl Zeiss, Inc, Germany) using a high-speed CMOS camera (Mikrotron Eosens 1362, Germany) at 250 Hz. Behavior image acquisition and stimulus projection was synchronized by the software controlling the behavior camera (Piper, Stanford Photonics).

## Imaging calcium activity induced by a live paramecium

Transgenic *Tg(NeuroD:GCaMP6f)* fish were embedded in agar and were placed under a one-photon spinning disc confocal microscope to acclimate for 10 min. They further acclimated for one minute with the 488 nm laser light on continuously before the onset of image acquisition to avoid detecting the strong initial activation of visual response in response to light onset. The laser was on continuously throughout the acquisition session to avoid distracting the animal with flashing light. We limited our imaging to a single plane that contained the pretectal area around AF7 (*Semmelhack et al., 2014*), recording at 5x magnification (0.25NA, air objective, Zeiss Fluar) at 13–15% laser power, with an output laser light at the objective of 150 µW/cm$^2$, and acquisition frequency at 3.6 Hz. The x/y optical resolution of the microscope used was 5.4 µm / pixel. Spontaneous activity was recorded for 1500 frames (about 7 min). A single paramecium was then added to a small well cut out in the agar in front of the fish (*Muto et al., 2013*). The well was sealed with a small lid of agar to keep the paramecium in front of the fish and avoid evaporation. Brain activity in response to the paramecium was recorded for 2500 frames (or 11.6 min). A Logitech C525 webcam (Logitech, USA) was placed under the fish to film the position of the paramecium using dark field illumination with an IR light source. A uEye CCD camera (IDS Imaging Development Systems GmbH) was attached to the microscope side port to record eye position. A notch filter 488 nm (Chroma, USA) was placed in front of the webcam to block out the imaging laser light, a 488 band pass filter (Chroma, USA) was used to image GCaMP6f fluorescence and a dichroic mirror (T470lpxr, Chroma, USA) reflected wavelengths below 470 nm and above 750 nm to the uEye camera while transmitting green photons to the fluorescence camera. The webcam and the uEye camera were controlled by custom-written software written in MATLAB so frames were acquired every time a fluorescence frame was acquired. Acquisition was synchronized by a TTL pulse sent from the fluorescence imaging software Slidebook (3I, USA) to MATLAB. It has recently been suggested that the light intensities used for one photon light-sheet microscopy stimulate the blue and UV cones of the retina which compromises visual perception (*Wolf, 2015*). At 5x magnification, light intensity at the focal plane was 50-75µW, which is substantially less than intensities used for light-sheet microscopy. Low magnification imaging resulted some scattered blue light that we supplemented with visible blue LED side illumination, for the fish to see the paramecium in front of it and maximize responses. Data analysis is described in Materials and methods, Behavioral data analysis and statistics and Calcium and behavior imaging data preprocessing.

## Chemical ablation and free-swimming prey capture assay

We crossed *Tg(gng8:Gal4;UAS:GFP)* fish with *Tg(UAS:NTR-mCherry)* fish. At day 4 dpf, we screened for expression of nitroreductase using mCherry fluorescence, excluded fish also expressing GFP, and selected two groups of healthy fish: ones that were positive and ones that were negative for NTR. We used the same feeding protocol described above on 5 and 6 dpf. On the evening of day six, we split the NTR positive and negative fish into two groups each and incubated them on a 48 well plate with either 5 mM MTZ (Sigma-Aldrich) in 0.2% DMSO (Sigma-Aldrich), referred to as +MTZ group, or 0.2% DMSO only (-MTZ group), overnight for 19 hr. The next morning, we washed the fish three times (five minutes per wash) with E3 fish water and placed them in the incubator to recover for 1 hr. After this period of recovery, fish were placed in a clear plastic 9-well plate (concave wells, diameter 11 mm, depth 10 mm) to record post-treatment feeding behavior. For our behavioral assay of predation, we placed the fish in the plate alternating in neighboring wells between the four groups (in a sequence of: NTR+/+MTZ, NTR+/-MTZ, NTR-/+MTZ, NTR-/-MTZ, and then repeat and so on). Fish behavior was recorded continuously for the first 30 s out of each 60 s period. The observation period consisted of an initial 10 min baseline period and then 20 more minutes after paramecia were added. Fish behavior was imaged with a uEye CCD camera (IDS Imaging Development Systems GmbH) at

10 Hz frame rate, with IR illumination and incidental room light. Paramecia were added in a 200 µL volume of pre-counted paramecia. The number of paramecia at start was on average 29.7 ± 8.9 (SD) and did not significantly differ between groups (*Figure 7—figure supplement 1D*). Eye convergence was measured manually by counting the frequency and number of frames that the eyes were converged. Paramecia were counted in three 15-frame long windows per time point, (at frames 100–115, 200–215 and 285–300) and averaged to obtain the number of paramecia at each time point. Swim speed was measured by tracking the fish's location in the well by using a custom MATLAB (Mathworks, USA) script. For all analysis, the experimenter was blind to the genetics and treatment of the animals.

## Fluorescence analysis

After the behavioral experiment, NTR+ fish were fixed in 4% formaldehyde overnight and then washed and mounted in low-melting agarose. Z-stacks of both habenulae were taken on a Zeiss LSM 880 upright laser scanning confocal at equal laser intensity, making sure no pixels were oversaturated. The total volume of expressing cells was quantified by using the surface function in Imaris microscopy image analysis software (Bitplane AG, Switzerland).

## Behavioral data analysis and statistics

All data was analyzed using custom-written software in MATLAB unless otherwise indicated. All pairwise comparisons were made with two-sided permutation tests using the difference in means as a test statistic (*Good, 2005*) unless otherwise indicated. Permutation tests do not make any assumptions about the underlying distribution, do not require equal variances or equal sample size. We rearranged labels (i.e. prey-experienced or prey-naïve) on observed data points and calculated the new test statistic 100,000 times thus creating a null distribution (under the null hypothesis, labels are interchangeable). We then computed the p-value by calculating the probability of obtaining a test statistic with an absolute value at least as great as the absolute value of the observed statistic (the difference in means between the actual experimental groups) under the null distribution. Permutation tests were two-tailed, because of the comparison of absolute values. We used a significance level $\alpha = 0.05$ (or a 5% chance of incorrectly rejecting the null hypothesis). Raw data from individual fish is plotted along with a boxplot summarizing the distribution statistics of the group: the central bar is the median, the bottom and top edges of the box indicate the 25th and 75th percentiles, respectively, and the whiskers extend to the most extreme data points not considered as statistical outliers (as defined by the inbuilt 'boxplot.m' function). We used a bootstrapping technique to calculate the percent increase in eye convergence frequency (*Figure 2F*).

Behavioral units of the prey capture sequence (pursuits and successful captures) were counted manually (*Figure 1*). Swimming velocity and percent of time resting were determined using custom tracking code. In the virtual environment setup, eye convergence events and tail flicks were also detected manually. A hit was defined as an eye convergence event when a stimulus was presented and a false alarm was an eye convergence when a blank stimulus was presented. Hit rate was defined as the number of hits divided by the number of stimulus trials and false alarm rate was estimated as the number of false alarms by blank stimulus trials. Blank stimuli were interleaved with stimulus trials throughout the experiment so we used the same false alarm rate for all contrast levels. To measure eye convergence rates compared to baseline for a given fish at each contrast level, we calculated the discriminability index $d'=Z(hit\ rate) -Z(false\ alarm\ rate)$, where Z is the inverse of the cumulative Gaussian distribution. d' distributions for prey-experienced and prey-naïve fish were compared using a two-way ANOVA test.

## Calcium and behavior imaging data pre-processing

Movies were registered using rigid body transformation (dftregistration from the MATLAB File Exchange). Regions of interest (ROI) were drawn manually around the left and right telencephalon, habenula, pretectum, tectal neuropil, tectal PVNs, cerebellum, parallel fibers of the crista cerebellaris (*Bae et al., 2009*) and the hindbrain. Left and right regions that appeared to have symmetrical activity around events of interest were averaged. Images were bleach corrected by fitting a single or double exponential (depending on the best goodness-of-fit) to the mean baseline fluorescence of each ROI excluding outliers and subtracting it from each pixel's fluorescence time series. Fluorescence

time-series of each pixel was then z-scored by subtracting the mean of the whole signal and dividing by the standard deviation (all Fz units in standard deviations away from the mean). We further corrected the average fluorescence traces of our ROIs for motion artifacts induced by body movements by interpolating values for all frames that had a displacement larger than two pixels. For *Figures 5* and *6*, we averaged left and right fluorescence for where signals were similar on either side. For analysis of fluorescence around pretectum transients (*Figure 6*) we detected pretectal peaks by thresholding the traces at two standard deviations from the mean and correcting for any aberrations manually and considered the highest value after thresholding as the peak. For each transient we considered the contralateral pretectum (to the prey) to be the side with the highest amplitude at peak time. We could not use prey position to determine contralateral identity because the prey was often on the midline and both eyes could detect it, and the prey moved faster than calcium transients rose to maximum meaning that a prey might evoked a transient while on one side of the fish, and already be on the other side by the time the transient has reached its peak. We extracted the average fluorescence in pretectum, optic tectum, cerebellum and hindbrain around 30 frames (~8 s) before and 30 frames after each pretectal peak. We calculated a baseline for each trace by averaging the fluorescence of the first 13 frames (~3.5 s).

To detect eye angle, for each frame eye contours were identified using custom-written software, and an ellipse was fit to the contours. Eye angle was considered to be the angle of the major axis of the ellipse relative to the midline of the fish. Eye vergence was the angle between the two eyes. Eye convergences were detected semi-automatically by identifying frames where both eye moved sharply towards the midline, thresholding vergence at 30° (*Bianco et al., 2011*), and correcting any aberrant detections manually. Speed of acquisition did not enable us to track fast changes in tail angle, so tail movements were detected by calculating pixel intensity changes on either side of the tail, subtracting a baseline rolling average over 20 frames. Pixel intensity changes matched tail bend amplitudes remarkably well when we scored movies by eye. Tail flick time points were detected semi-automatically by thresholding at two standard deviations from the mean and corrected for aberrations.

## Quantifying prey position for the encoding model

We preprocessed prey-position movies by subtracting a baseline rolling average over 20 frames. Prey position was quantified using a polar representation of space around the fish. We defined 19 angle bins (or angle basis functions) of prey position relative to the fish's midline. Each bin was represented by a von Mises distribution (which is an approximation of the circular normal distribution) with centers evenly spaced from –pi to pi and the width parameter kappa set to 20. When the prey was close to the center of a bin, that angle was weighted strongly, whereas when the prey was in between two bins, the angles at the centers of those bins were weighted equally, generating a more continuous representation of prey-space. Similarly, we also defined five radial basis functions, which describe distance of the prey from the fish, using Gaussian distributions with centers evenly spaced from 0 to ~5 mm and width parameter sigma set to 40. We then took the product of each angle basis function and each radial basis function, yielding a total of 95 two-dimensional spatial basis functions that vary in both angle and radius. Each frame of the prey video was then projected onto each of these basis functions, producing 95 prey-location time series. For a particular prey-location time series the value is high for times when the prey is near the specified angle and radius and zero at other times.

## Pixel-wise encoding model estimation and validation

Linearized finite impulse response (FIR) encoding models (*Nishimoto et al., 2011*; *Huth et al., 2016*) that predict pixel fluorescence based on the location of prey were estimated for each pixel in each fish. For each pixel (1) we constructed the stimulus input matrix that represents prey location over time: To account for calcium indicator kinetics, and neural response delays relative to movement of the prey, each of the 95 prey-location time series were delayed from −5 to + 15 frames (−1.4 to 4.2 s), yielding a total of 1900 features that were used to predict pixel fluorescence. The shifted prey-location time series were concatenated, and the mean of each 1900 feature time series was then subtracted to avoid fitting an intercept term in the regression. (2) We used ridge regression to estimate the model coefficients that quantify the relationship between prey location and pixel

fluorescence. To enable unbiased assessment of model prediction performance we used a 10-fold cross-validation (the outer-layer validation) approach to fit and validate the encoding models. First, the full dataset for each fish was divided into 10 sequential temporal segments. For each fold, one segment was reserved for model validation and the other nine segments were used to estimate the model weights by way of L2-regularized linear regression (ridge regression). (3) We estimated the regularization parameter for each of the 10 outer-layer folds using a second lever of cross-validation. We tested 20 regularization parameters α, log spaced between 1 and 1000. For each parameter α, the following procedure was repeated 50 times: we randomly selected and removed 400 time points (10 blocks of 40 consecutive time points each) from the model estimation dataset. Model weights were then estimated using the remaining time points and used to predict responses in the 400 selected time points. After this procedure was repeated 50 times a regularization-performance curve was obtained for each outer-fold layer by averaging the 50 prediction performance values for each regularization parameter. The regularization parameter with the best prediction performance was selected. (4) We re-computed model weights using the entire model estimation dataset (consisting of the nine segments of data for this cross-validation fold). (5) We predicted fluorescence for the held-out segment of data using the estimated weights. Both weights and predicted fluorescence were saved. (6) We repeated steps 3 to 5 for each of the 10 outer-layer cross-validation folds. (7) After all 10 folds had been completed the predicted segments were concatenated to form a complete prediction dataset of the same size as the original fluorescence data. The Pearson correlation between the complete predicted time series and actual fluorescence time series was then computed for each pixel. For further analysis of pixel selectivity (8) we averaged together the estimated model weights from each of the 10 cross-validation folds (average across delays to obtain spatial receptive fields from *Figure 3C*, average across radii to obtain preferred angle, or average across radii and angles to obtain preferred delay, etc). (9) Statistical significance of predictions was computed by comparing estimated correlations to the null distribution of correlations between two independent Gaussian random variables of the same length. Resulting p-values were corrected for multiple comparisons within each fish using the false discovery rate (FDR, q < 0.05) procedure (*Benjamini and Hochberg, 1995*).

All model fitting was performed using custom software written in Python (https://github.com/alexhuth/ridge; *Huth, 2020*; copy archived at https://github.com/elifesciences-publications/ridge).

## Granger-causality from calcium fluorescence imaging data

On the basis of our previous study (*Fallani et al., 2015*), we studied Granger-causality between neuronal GCaMP6f fluorescence signals with the framework described below. According to the concept of Granger-causality (*Granger, 1969*; *Bressler and Seth, 2011*), a variable $F_1$ causes $F_2$ ($F_1 \rightarrow F_2$) if the prediction of $F_2$ is improved when information from $F_1$ is included in the prediction model for $F_2$. GC measure is typically based on autoregressive (AR) models. In a bivariate AR modeling, a stationary signal $x_2(t)$ can be expressed as a linear regression of its past values according to the formula:

$$F_2(t) = \sum_{k=1}^{q} a(k) F_2(t-k) + e_1(t) \tag{1}$$

where $a(k)$ are the regression coefficients of the univariate AR model, $q$ is the model order, and $e_1(t)$ is the respective prediction error. By introducing the information from the stationary signal $F_1(t)$, the formula can be rewritten as:

$$F_2(t) = \sum_{k=1}^{q} b_2(k) F_2(t-k) + \sum_{k=1}^{q} b_1(k) F_1(t-k) + e_2(t) \tag{2}$$

where $b_1(k)$ and $b_2(k)$ are the new regression coefficients of the bivariate AR model, and $e_2(t)$ is the new prediction error obtained by including also the past of $F_1(t)$ in the linear regression of $F_2(t)$. Statistical influence (Granger causality) between $F_1(t)$ and $F_2(t)$ is evaluated by the log ratio of the prediction error variances for the bivariate and univariate model:

$$GC_{1 \rightarrow 2} = ln\left(\frac{var[e_2(t)]}{var[e_1(t)]}\right) \tag{3}$$

By construction, GC is a positive number; the higher $GCI_{1\to2}$, the stronger the influence of $F_1(t)$ on $F_2(t)$ is. Such influence is often considered to reflect the existence of an information flow outgoing from the system $F_1(t)$ towards the system $F_2(t)$ (**Granger, 1969**; **Bressler and Seth, 2011**). Finally, GC is generally an asymmetric measure (i.e. $GCI_{1\to2} \neq GCI_{2\to1}$), which allows inferring causal or driver-response relationships.

The regression coefficients of the AR models were computed according to the ordinary-least-squares minimization of the Yule-Walker equations (**Bressler and Seth, 2011**; **Gourévitch et al., 2006**). The model order $q$ was selected according to the Akaike criterion (**Akaike, 1974**). This criterion finds the optimal $q$ that minimizes the following cost function $C(q) = \mathrm{Tln}(det(\Sigma_2)) + \frac{T(TN+qN^2)}{T-qN-N-1}$ N = 2, where $\Sigma_2$ N = 2 is the noise covariance matrix of the bivariate AR model, *N = 2*, and *T* is the number of samples of the time series. Basically, this cost function balances the variance accounted for by the AR model against the number of coefficients to be estimated. We fixed the common model order of q = 5 frames for all fish (mean of optimal order values obtained for individual fish) (**Pereda et al., 2005**).

We estimated GC between GCaMP6f fluorescence signals in spontaneous (without prey) and evoked (with prey) conditions over 1500 frames (7 min) and 2500 frames (11 min) respectively (normalized to zero mean and unitary variance). Each zebrafish's brain was thus characterized by a full connectivity pattern by quantifying GC influences between identified ROIs. The strength of a functional link between two regions was estimated with the value of the F-statistic, which quantifies the statistical significance of the directed interaction under the assumption of non-directed effect. Statistical differences between groups were evaluated using a Welsh-test for each link (**Bressler and Seth, 2011**; **Gourévitch et al., 2006**). Only p-values corresponding to percentiles inferior to a statistical threshold of $\alpha = 0.05$ FDR-BH corrected for multiple comparisons were retained.

Robust multivariate linear regression to relate prey capture initiation to Granger-causality links. We used multivariate linear regression to model the relationship between prey capture initiation frequency (response variable), and Granger-causality links strength and experience (two predictor variables). We used the 'robust' option in the MATLAB *fitlm* function, which reiteratively weights each data point to reduce the effect of outlier response points on the fit. A bisquare function was used for re-weighting.

## Acknowledgements

We thank Dr. Marnie Halpern for generously providing the Tg(gng8:Gal4) fish line; Kait Kliman, Mel Boren, Sonia Castillo, and Allison Kepple for maintaining fish lines at UC Berkeley; Natalia Maties, Bodgan Buzurin, and Sophie Nunes Figueiredo for maintaining transgenic lines at the ICM zebrafish facility; Holly Aaron, Jen Lee, and Feather Yves from the Berkeley Molecular Imaging Center at UC Berkeley. Intelligent Imaging Innovations provided valuable advice and assistance with the optical system. We thank Adna Dumitrescu for developing approaches to test prey consumption and Amy Winans, Carlos Pantoja, and Shih-Wei Chou for thoughtful feedback on the manuscript. We thank Vincent Guillemot from the ICM Biostatistics/Bioinformatics core facility, and Chris Holdgraf from UC Berkeley for providing advice on statistics. We also thank UC Berkeley undergraduate researchers Andrea Romo, Sweta Parija and Lydia Liu for help with data analysis. This research was developed with funding from the Defense Advanced Research Projects Agency (DARPA), Contract No. N66001-17-C-4015 to JG and EYI, with support from the National Institutes of Health Nanomedicine Development Center for the Optical Control of Biological Function (2PN2EY018241) to EYI, an Optoloco starting grant from ERC (#311673) to CW, a Research-in-Paris postdoctoral fellowship and an EMBO postdoctoral fellowship (# ALTF 549–2013) to AP, and a Boehringer Ingelheim Fonds fellowship and Chateaubriand graduate student fellowship to CSO.

## Additional information

### Competing interests

Claire Wyart: Reviewing editor, *eLife*. The other authors declare that no competing interests exist.

## Funding

| Funder | Grant reference number | Author |
| --- | --- | --- |
| Defense Advanced Research Projects Agency | N66001-17-C-4015 | Jack L Gallant Ehud Y Isacoff |
| National Institutes of Health | 2PN2EY018241 | Ehud Y Isacoff |
| ERC | #311673 | Claire Wyart |
| EMBO | postdoctoral fellowship (# ALTF 549–2013) | Andrew Prendergast |
| Boehringer Ingelheim Fonds | PhD fellowships | Claire S Oldfield |
| Embassy of France | Chateaubriand graduate student fellowship | Claire S Oldfield |

The funders had no role in study design, data collection and interpretation, or the decision to submit the work for publication.

## Author contributions

Claire S Oldfield, Conceptualization, Software, Formal analysis, Investigation, Writing - original draft; Irene Grossrubatscher, Conceptualization, Software, Formal analysis, Investigation, Methodology, Writing - original draft, Writing - review and editing; Mario Chávez, Adam Hoagland, Alex R Huth, Software, Formal analysis; Elizabeth C Carroll, Methodology; Andrew Prendergast, Formal analysis; Tony Qu, Resources; Jack L Gallant, Conceptualization, Supervision; Claire Wyart, Conceptualization, Supervision, Writing - review and editing; Ehud Y Isacoff, Conceptualization, Resources, Formal analysis, Supervision, Writing - original draft, Project administration, Writing - review and editing

## Author ORCIDs

Irene Grossrubatscher (iD) https://orcid.org/0000-0001-7285-6375
Jack L Gallant (iD) http://orcid.org/0000-0001-7273-1054
Claire Wyart (iD) http://orcid.org/0000-0002-1668-4975
Ehud Y Isacoff (iD) https://orcid.org/0000-0003-4775-9359

## Ethics

Animal experimentation: This study was performed in strict accordance with the recommendations in the Guide for the Care and Use of Laboratory Animals of the National Institutes of Health. All of the animals were handled according to approved institutional Animal Care and Use Committee (ACUC) of the University of California, Berkeley. Protocol ID: AUP-2015-06-7705-1, last approval date (11/20/2019).

## Decision letter and Author response

Decision letter https://doi.org/10.7554/eLife.56619.sa1
Author response https://doi.org/10.7554/eLife.56619.sa2

# Additional files

## Supplementary files

• Supplementary file 1. P-values for permutation test comparison of average fluorescence trace before and after eye convergence in experienced fish for different brain regions (*Figure 2H*).

• Supplementary file 2. P-values for comparisons of Granger-causality links in the visual areas (matrices presented in *Figure 4*). For each link, differences between experienced and naïve fish are assessed. Threshold for significance after FDR-BH correction is p=0.0017 (see Materials and methods).

• Supplementary file 3. Statistical comparisons of visual area - motor area interactions in experienced versus naïve fish and strongest versus weakest hunters among experienced fish. P-values for comparisons of directed (causal) interactions between visual and motor areas between experienced and

naïve fish (matrices presented in *Figure 5A*), and strongest and weakest hunters among experienced fish (matrices presented in *Figure 5—figure supplement 1A*). Threshold for significance after FDR-BH correction is top: p=3.E-04, and bottom p=0.003. Links that are significantly different between groups are highlighted in yellow.

- Transparent reporting form

### Data availability

All data generated or analysed during this study are included in the manuscript and supporting files. Source data files have been provided for all figures.

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
