## [Decision Letter]

**Acceptance summary:**

In this manuscript, the authors looked at prey capture behaviors (eye convergence and tail flick) of zebrafish larvae in a virtual reality environment while at the same time they observed the neuronal activity of the brain. Importantly, they focused on the effect of previous experience on the prey capture performance and its neuronal correlates. They compared fish fed live paramecia (experienced) and fish fed fish flakes (naïve), the former of which showed higher response to live paramecia. As a result, authors found the correlate of learning-dependent improvement of prey capture in the brain, and propose a new idea of forebrain involvement in the reflex-like behavior in larval zebrafish. All three major concerns by the reviewers were addressed by additional experiments and textual revisions, supporting and strengthening authors' conclusions.

**Decision letter after peer review:**

Thank you for submitting your article "Experience, circuit dynamics and forebrain recruitment in larval zebrafish prey capture" for consideration by *eLife*. Your article has been reviewed by three peer reviewers, and the evaluation has been overseen by a Reviewing Editor and K VijayRaghavan as the Senior Editor. The reviewers have opted to remain anonymous.

The reviewers have discussed the reviews with one another and the Reviewing Editor has drafted this decision to help you prepare a revised submission.

Summary:

In this manuscript, the authors looked at prey capture behaviors (eye convergence and tail flick) of zebrafish larvae in a virtual reality environment while at the same time they observed the neuronal activity of the brain in a fixed focal plane. Importantly, they focused on the effect of previous experience on the prey capture performance and its neuronal correlates. They compared fish fed live paramecia (experienced) and fish fed fish flakes (naïve).

They first looked at the behavior and found that eye convergence of experienced fish in response to prey was higher than in naïve fish (Figure 1). They then monitored the brain and found that the activity of the visual areas, pretectum and tectum, are not very different between experienced and naïve fish. There was also no difference between neuronal activities at eye convergence events that spontaneously occurred without prey and those at prey-evoked eye convergence (Figure 2, Figure 2—figure supplement 1). To see whether encoding of visual stimuli is different, they performed regularized regression analysis on the neuronal map in the pretectum, tectal neuropil and tectal PVN, and found no difference between experienced and naïve fish (Figure 3). To look at information flow in and out of visual areas, they performed Granger causality analyses. Granger causality between tectum and PVN was increased by presentation of prey, but again, prior experience made no difference (Figure 4). On the other hand, the authors did observe a difference in information flow to the telencephalon from visual areas. Interestingly, looking at individual fish, eye convergence performance was correlated with Granger causality from pretectum and other visual areas, and from visual areas to the telencephalon. This raised the possibility that the forebrain may be involved in increased performance resulting from experience (Figure 5). Triggered by this observation, the authors looked at activity of each area at the time of pretectal activity peaks, and found that responses of the telencephalon and habenular region were different between experienced and naïve fish (Figure 6). To confirm the relevance of brain activity measurements in the habenulae, they performed a functional test to kill habenular neurons using a toxin, and indeed saw decreased success at prey capture (Figure 7).

Essential revisions:

The writing and figures are generally very good, and most of the small-scale interpretations are sound. However, it is the consensus of the reviewers that the following concerns need to be addressed before the study is publishable.

1) There is a simple and appealing (maybe preferable) alternative explanation for the results: flake-fed larvae are more satiated than paramecium-fed larvae are. We acknowledge the great lengths that the authors have gone to in demonstrating that larvae in the two groups are similar in size, move similarly, and have had access to saturating amounts of food. This does not mean that they are equally hungry (flakes are easier to catch and may be more nutritious, as this is what they're designed for). The behavioral effect, an increased initiation of pursuit, appears to be more a function of motivation than hunting proficiency. The authors have demonstrated that the visual coding does not depend on experience, nor is there evidence of improved efficacy (say, improved motor coordination) once an initiation occurs. These seem like more motivated animals. Greater hunger is the simplest explanation, and the authors capably outline in their Discussion the mechanisms by which hunger could mediate this stronger motivation to prey.

We do not think that the authors' model, "experience leads to greater visually-evoked activity in the forebrain", is supported until the effects of hunger and experience are uncoupled. Unfed fish, which would undoubtedly be hungry and naïve, would provide the answer. If they initiate more and show other effects reported in the manuscript, then it is likely that hunger is responsible. If they act more like flake-fed larvae (or are even farther in this direction), then the authors' model has much stronger support. While it would be less than ideal, and would leave caveats, a clear behavioral effect (even in the absence of brain imaging) could be used to provide this support.

2) The habenular ablation experiments support that the habenulae are necessary for normal initiation and capture. This is only circumstantially related to the manuscript's other findings. The solution is to do the ablations of flake-fed larvae, where the model would predict no effect.

3) Subsection “Fluorescence Analysis”. The description of quantifying habenular neuronal loss is confusing. The authors state that "NTR+ fish were fixed in 4% formaldehyde overnight and then washed and mounted in low-melting agarose. Z-stacks of both habenulae were taken on a Zeiss LSM 880 upright laser scanning confocal at equal laser intensity, making sure no pixels were oversaturated. The total volume of expressing cells was quantified.…." However, it is well known that fixation in paraformaldehyde quenches the fluorescence signal. Please clarify how the authors were able to use this signal (mCherry?) to quantify neuronal loss.

---

## [Author Response]

Essential revisions:The writing and figures are generally very good, and most of the small-scale interpretations are sound. However, it is the consensus of the reviewers that the following concerns need to be addressed before the study is publishable.1) There is a simple and appealing (maybe preferable) alternative explanation for the results: flake-fed larvae are more satiated than paramecium-fed larvae are. We acknowledge the great lengths that the authors have gone to in demonstrating that larvae in the two groups are similar in size, move similarly, and have had access to saturating amounts of food. This does not mean that they are equally hungry (flakes are easier to catch and may be more nutritious, as this is what they're designed for). The behavioral effect, an increased initiation of pursuit, appears to be more a function of motivation than hunting proficiency. The authors have demonstrated that the visual coding does not depend on experience, nor is there evidence of improved efficacy (say, improved motor coordination) once an initiation occurs. These seem like more motivated animals. Greater hunger is the simplest explanation, and the authors capably outline in their Discussion the mechanisms by which hunger could mediate this stronger motivation to prey.We do not think that the authors' model, "experience leads to greater visually-evoked activity in the forebrain", is supported until the effects of hunger and experience are uncoupled. Unfed fish, which would undoubtedly be hungry and naïve, would provide the answer. If they initiate more and show other effects reported in the manuscript, then it is likely that hunger is responsible. If they act more like flake-fed larvae (or are even farther in this direction), then the authors' model has much stronger support. While it would be less than ideal, and would leave caveats, a clear behavioral effect (even in the absence of brain imaging) could be used to provide this support.

To address this concern, we performed a new experiment. We fed sibling fish either paramecia or flakes on day 5 and 6 dpf ad libitum, and then starved the fish overnight, as in our usual protocol. On day 7, when we normally perform our assays, we performed a flake-feeding test. In this test, we let fish feed on flakes for 10 minutes. We assessed the number of eye convergences during that time as a proxy for the motivation to hunt (observation of food ingestion was not possible to do reliably). If the flake fed fish are less hungry, they would display fewer eye convergences. Instead, we observed a significantly higher number of eye convergences in response to the flakes in the previously flakes-fed fish. This shows that flake-trained fish are not less hungry. Moreover, it suggests that the experience with flakes boosts flake “hunting” in the same way as experience with paramecia boosts paramecia hunting, supporting our conclusion that experience improves performance and demonstrating that this effect is specific to the identity of the food. (subsection “Prior experience of prey increases prey capture initiation in larval zebrafish”; Figure 1, Figure 1—figure supplement 1G).

We further tested the interpretation that experience with a particular food source enhances feeding/hunting of that food source. We did this in an experiment with another live prey. Fish were fed with rotifers *ad libitum* in two groups. Group 1 was fed for one day (at 5 dpf) and group 2 was fed for two days (4 and 5 dpf), both groups were starved overnight, until day 6. On day 6, both groups were given fluorescent rotifers for 30 minutes. We fixed the fish and compared the levels of fluorescence in their digestive tract. We observed that fish that were exposed to rotifers for two days consumed more fluorescent rotifers than fish that were fed on only one day (Author response image 1). This observation supports the conclusion that experience with prey is the dominating factor. The number of fish we tested was small (N = 10 fish in each group) and the therefore can only show a trend. Power analysis reveals that it would take N = 80 fish to reach significance. We have not been able to obtain sufficient numbers (N = 160) of sibling fish to make this possible. We therefore present this result only for the review.

These experiments support our interpretation that experience and not hunger is the driving force of improvement in prey capture.

**Author response image 1. respfig1:** Rotifer consumption quantification in larvae with different food experience levels. (**A**) Example images of gut food fluorescence signal (magenta) from animals fed live rotifers stained with TexasRed at 6 dpf for 30 minutes. We analysed data from larvae which were previously introduced to live prey (unstained rotifers) at 5 dpf (1 day food experience) versus more experienced animals fed at 4&5 dpf (2 days food experience). Bottom pannels display TexasRed signal, dotted lines demarcate the gut area used to quantify fluorescence intensity. (**B**) Gut fluorescence quantification of food ingested in a 30 minute period. Coloured symbols denote the fluorescence level from example images. Error bars represent means ± SD.

2) The habenular ablation experiments support that the habenulae are necessary for normal initiation and capture. This is only circumstantially related to the manuscript's other findings. The solution is to do the ablations of flake-fed larvae, where the model would predict no effect.

We repeated our paramecium consumption experiment with flakes-fed sibling fish that were or were not subjected to habenula chemo-ablation. NTR+ fish were fed on flakes on days 5 and 6 and then, overnight between days 6 and 7, given either DMSO alone (-MTZ) or DMSO + MTZ (+MTZ). They were then tested for feeding on paramecia on day 7. There was no statistical difference between the +MTZ, which had habenular damage, and the -MTZ control (indeed, +MTZ consumed slightly, but not significantly, more). This result indicates that the habenula is not necessary for normal initiation and capture. (Subsection “Forebrain disruption reduces hunting initiation in prey-experienced fish”, Figure 7—figure supplement 1G).

3) Subsection “Fluorescence Analysis”. The description of quantifying habenular neuronal loss is confusing. The authors state that "NTR+ fish were fixed in 4% formaldehyde overnight and then washed and mounted in low melting agarose. Z-stacks of both habenulae were taken on a Zeiss LSM 880 upright laser scanning confocal at equal laser intensity, making sure no pixels were oversaturated. The total volume of expressing cells was quantified.…." However, it is well known that fixation in paraformaldehyde quenches the fluorescence signal. Please clarify how the authors were able to use this signal (mCherry?) to quantify neuronal loss.

Fixation with paraformaldehyde can indeed quench fluorescent protein signal. Under our conditions, the mCherry signal remained bright and we were careful to fix all of the animals in each experiment simultaneously, for the same time, and in the same solutions. Differences in signal between animals more likely reflect typical animal to animal variation in transgene expression. We observed a range of mCherry brightness in both +MTZ and -MTZ sibling NTR+ fish and clearly distinct relations of brightness to volume between the two groups (as we show in new Figure 7—figure supplement 2B), supporting the accuracy of the volume measure in Figure 7C an accurate representation of chemo-ablation. Moreover, across animals with different levels of mCherry signal, we see that the -MTZ controls have a typical diffuse pattern throughout the habenula, whereas the +MTZ-treated animals have a blotchy distribution (Figure 7B and Figure 7—figure supplement 2A), which is typical of chemo-ablation. We have added more example images of +MTZ and -MTZ fish to illustrate this (Figure 7—figure supplement 2A) and we have added a description of this to the figure.